# Epigenetic memory is governed by an effector recruitment specificity toggle in Heterochromatin Protein 1

Amanda Ames [1], Melissa Seman[1,6], Ajay Larkin [1,6], Gulzhan Raiymbek[2], Ziyuan Chen[3], Alex Levashkevich[1], Bokyung Kim[4], Julie Suzanne Biteen [3,5] & Kaushik Ragunathan [1] ✉

HP1 proteins are essential for establishing and maintaining transcriptionally silent heterochromatin. They dimerize, forming a binding interface to recruit diverse chromatin-associated factors. Although HP1 proteins are known to rapidly evolve, the extent of variation required to achieve functional specialization is unknown. To investigate how changes in amino acid sequence impacts heterochromatin formation, we performed a targeted mutagenesis screen of the *S. pombe* HP1 homolog, Swi6. Substitutions within an auxiliary surface adjacent to the HP1 dimerization interface produce Swi6 variants with divergent maintenance properties. Remarkably, substitutions at a single amino acid position lead to the persistent gain or loss of epigenetic inheritance. These substitutions increase Swi6 chromatin occupancy in vivo and altered Swi6-protein interactions that reprogram H3K9me maintenance. We show how relatively minor changes in Swi6 amino acid composition in an auxiliary surface can lead to profound changes in epigenetic inheritance providing a redundant mechanism to evolve HP1-effector specificity.

Chromatin organization, which is critical for maintaining genome integrity and regulating gene expression, depends in part on the post-translational modifications of DNA packaging proteins called histones. These modifications enable the establishment of distinct chromatin compartments consisting of active or repressed genes, and their inheritance following DNA replication enables cells to maintain unique identities[1,2]. Disrupting the epigenetic landscape leads to aneuploidy and genome instability, two established hallmarks of cancer. Over 50% of sequenced cancers have at least one mutation in histones, histone-binding proteins, or nucleosome remodelers, underscoring the devastating impacts of epigenetic misregulation in cancer[3]. Elucidating fundamental mechanisms of epigenetic regulation is critical to deciphering how gene expression is regulated and potentially informs the development of innovative therapeutic strategies.

Establishment depends on the sequence-specific recruitment of histone modifiers to unique locations in the genome[4]. These modifications can be propagated over multiple cell divisions in a DNA sequence-independent manner through a process known as maintenance or epigenetic inheritance. The molecular basis for epigenetic inheritance is thought to involve a process called read-write, where pre-existing histone modifications recruit enzymes to modify newly incorporated histones after DNA replication[5–7]. However, several non-histone proteins that bind histone modifications also impact epigenetic inheritance by inducing changes in chromatin organization[8,9]. For example, in vitro reconstitution studies show that PRC1 (polycomb repressive complex 1), a protein complex involved in histone H3 lysine 27 methylation (H3K27me) dependent silencing, remains continuously bound to old and newly replicated DNA independent of H3K27me[10].

[1]Department of Biology, Brandeis University, Waltham, MA 02453, USA. [2]Department of Biological Chemistry, University of Michigan, Ann Arbor, MI 48109, USA. [3]Department of Biophysics, University of Michigan, Ann Arbor, MI 48104, USA. [4]Department of Biochemistry, Brandeis University, Waltham, MA 02453, USA. [5]Department of Chemistry, University of Michigan, Ann Arbor, MI 48104, USA. [6]These authors contributed equally: Melissa Seman, Ajay Larkin. ✉e-mail: kaushikr@brandeis.edu

Moreover, the ability of PRC1 to form condensates in conjunction with modified histones can tune epigenetic memory[11]. In an analogous silencing pathway, proteins called Heterochromatin Protein 1 (HP1) remain bound to histone H3 lysine 9 methylation (H3K9me), throughout DNA replication, suggesting HP1 proteins might be an integral part of the epigenetic imprint[12]. HP1 proteins also form condensates and oligomerize, which is thought to have a crucial role in maintaining epigenetic memory[13,14]. While the importance of these processes is well-recognized, the mechanisms that coordinate the interplay between histone and non-histone proteins to encode epigenetic memory remain poorly understood.

Canonical reader domains, such as bromodomains and chromodomains, are found in protein complexes that alter gene expression and chromatin accessibility[15]. Readers are enigmatic because, despite being highly conserved, their ability to be co-opted by diverse effectors enables them to fulfill functionally distinct roles in regulating gene expression[16]. HP1 proteins are evolutionary conserved reader domaining containing factors involved in heterochromatin formation[17]. HP1 proteins have a conserved chromodomain (CD), which binds specifically to H3K9me, and a chromoshadow domain (CSD) that promotes dimerization[18,19]. HP1 proteins can engage in higher-order interactions, leading to the formation of condensates that have liquid-like or gel-like properties in vitro and in vivo[20–22]. Upon dimerization, CSD subunits create a binding interface that facilitates protein interactions. HP1 binding partners contain variations of a consensus pentapeptide motif, with the defining feature being a central valine residue, such as the PxVxL motif in mouse CAF1[23,24]. Some HP1 variants demonstrate altered specificity for degenerate equivalents of this motif. Notably, the *S. pombe* HP1 paralog Chp2 binds to a chromatin remodeler Mit1 through a CkIvV motif[25]. HP1 proteins recruit factors that enhance and antagonize heterochromatin formation, many of which bind the same shared CSD interface[26]. This raises an important question about how the recruitment of factors with opposing activities is coordinated, given all these effectors presumably compete for the same binding interface.

HP1 proteins are among the most rapidly evolving protein families[27,28]. In Dipteran flies, phylogenomic studies reveal an unusually high number of HP1 gene duplications, leading to opportunities for functional specialization of paralogs. These young HP1 genes show an elevated percent identity at amino acid positions involved in H3K9me recognition and protein dimerization, indicating they are functional HP1 paralogs that evolved through positive selection[28]. Yet, even within these otherwise conserved domains, there is substantial variation in some amino acid positions within the CD and CSD that remain functionally undefined[23]. Non-conserved sequences may contribute to functional diversity in HP1 proteins by influencing some paralog-specific functions. In mouse HP1a, the N-terminal extension serves as a site for hyperphosphorylation, which leads to higher chromatin binding and compaction[29]. The CSD dimerization interface, influenced by amino acid composition in the CSD and C-terminal extension, differs across HP1 proteins and is thought to tune affinity for various protein ligands[30]. However, we have limited knowledge of how much sequence variation is required to alter the function of HP1 proteins and achieve functional specialization.

The fission yeast *Schizosacchromyces pombe* (*S. pombe*), represents an outstanding model system to study heterochromatin[31]. Heterochromatin establishment and maintenance involves H3K9me and the activity of an H3K9 methyltransferase in *S. pombe* called Clr4, a Suv39h homolog[32]. *S. pombe* has two HP1 reader proteins, Chp2 and Swi6, with extensive sequence and structural similarity to HP1 proteins in metazoans. Swi6 and Chp2 have distinct, additive roles in transcriptional silencing at pericentromeric repeats, telomeres, and the mating type locus[33]. Swi6 and Chp2 interact with distinct sets of heterochromatin regulators and have vastly different expression levels. Swi6 interacts with many factors, including those involved in RNAi-

mediated heterochromatin formation, the histone deacetylase (Clr3), and a putative H3K9 demethylase (Epe1)[26,34–37]. Chp2 recruits the Snf2/HDAC repressive effector complex (SHREC) that includes Clr3 and the chromatin remodeler Mit1 to promote transcriptional silencing[7,25,33,38]. Given the potential role of reader proteins, such as Swi6, in epigenetic inheritance, we anticipate that their plasticity to evolve new protein-protein interactions serves as a potential mechanism to tune epigenetic memory.

In this work, we determine how sequence changes in Swi6 influence epigenetic inheritance by performing a targeted PCR-based mutagenesis screen. By modifying the amino acid composition in the Swi6-CSD through a targeted, PCR-based mutagenesis strategy, we alter the durability of H3K9me-dependent epigenetic inheritance. Notably, we show substitutions at a single residue (Thr 278) are sufficient to achieve functional divergence. While some amino acid substitutions at this residue produce a gain of function maintenance phenotype, other substitutions led to a persistent loss of maintenance. Our study reveals an HP1 protein, and possibly other reader proteins alike, can display substantial plasticity wherein relatively minor variations in amino acid composition outside primary structural interfaces can produce strikingly different functional outcomes.

## Results

### Swi6 variants affect heterochromatin maintenance

To uncouple the effects of sequence-dependent heterochromatin establishment from epigenetic inheritance, we use a system where a TetR-Clr4-I fusion protein binds *10X-tetO* DNA binding sequences placed upstream of an *ade6+* reporter gene. TetR-Clr4-I binding promotes site-specific H3K9me deposition, leading to *ade6+* silencing[5,7]. Colonies appear red on low adenine media when *ade6+* is silenced and white when *ade6+* is expressed. The addition of tetracycline (+tet) triggers the release of TetR-Clr4-I, after which we can measure epigenetic inheritance in the absence of continuous initiation (Fig. 1a). Consistent with previous work, deleting the eraser of H3K9me, Epe1 (*epe1Δ*), promotes maintenance leading to the appearance of red and sectored colonies on +tetracycline-containing medium in contrast to white colonies in *epe1+* cells (Fig. 1b, *epe1+* versus *epe1Δ*, +tet)[7].

Next, we generated a site-directed mutagenesis library using tiling primers containing degenerate NNN codons targeting 65 amino acids of the Swi6-CSD domain given that it is the primary interaction site for heterochromatin-associated factors. This PCR-based *swi6*-CSD variant library was integrated into the *S. pombe* genome, replacing the endogenous *swi6+* sequence (Supplementary Fig. 1a). To identify Swi6 gain of function mutations that lead to enhanced maintenance, we transformed our *swi6*-CSD variant library in an *epe1+* background where cells are normally white when plated on +tet medium. We expected a gain of function Swi6 variant to produce red/sectored colonies on +tet medium. Based on an initial hit in our screen (T278Y), we discovered that several Thr 278 substitutions produce a gain of function maintenance phenotype, including phenylalanine (F), tyrosine (Y), alanine (A), cysteine (C), and serine (S) (Fig. 1b). This subset of residues are conserved or semi-conserved substitutions with uncharged side chains. When Thr 278 was replaced with amino acid substitutions containing charged side chains, we observed a gain of maintenance in the case of glutamate (E) and arginine (R) substitutions, but unexpectedly, we observed a loss of maintenance in the case of aspartate (D) and lysine (K) substitutions (Fig. 1c). These phenotypic differences were not due to changes in Swi6 protein levels, given the expression of all Thr 278 variants is comparable to Swi6-WT (Supplementary Fig. 1b, c).

We quantitatively measured transcriptional silencing of the reporter locus in *swi6* T278Y (*swi6-Y*) and *swi6* T278K (*swi6-K*). We performed quantitative real-time PCR (qRT-PCR) for a gene upstream of *10XTetO-ade6*, *SPCC330.06c*. Consistent with the *ade6+* silencing phenotype, *swi6-*Y, and *swi6-*K exhibited a similar decrease in

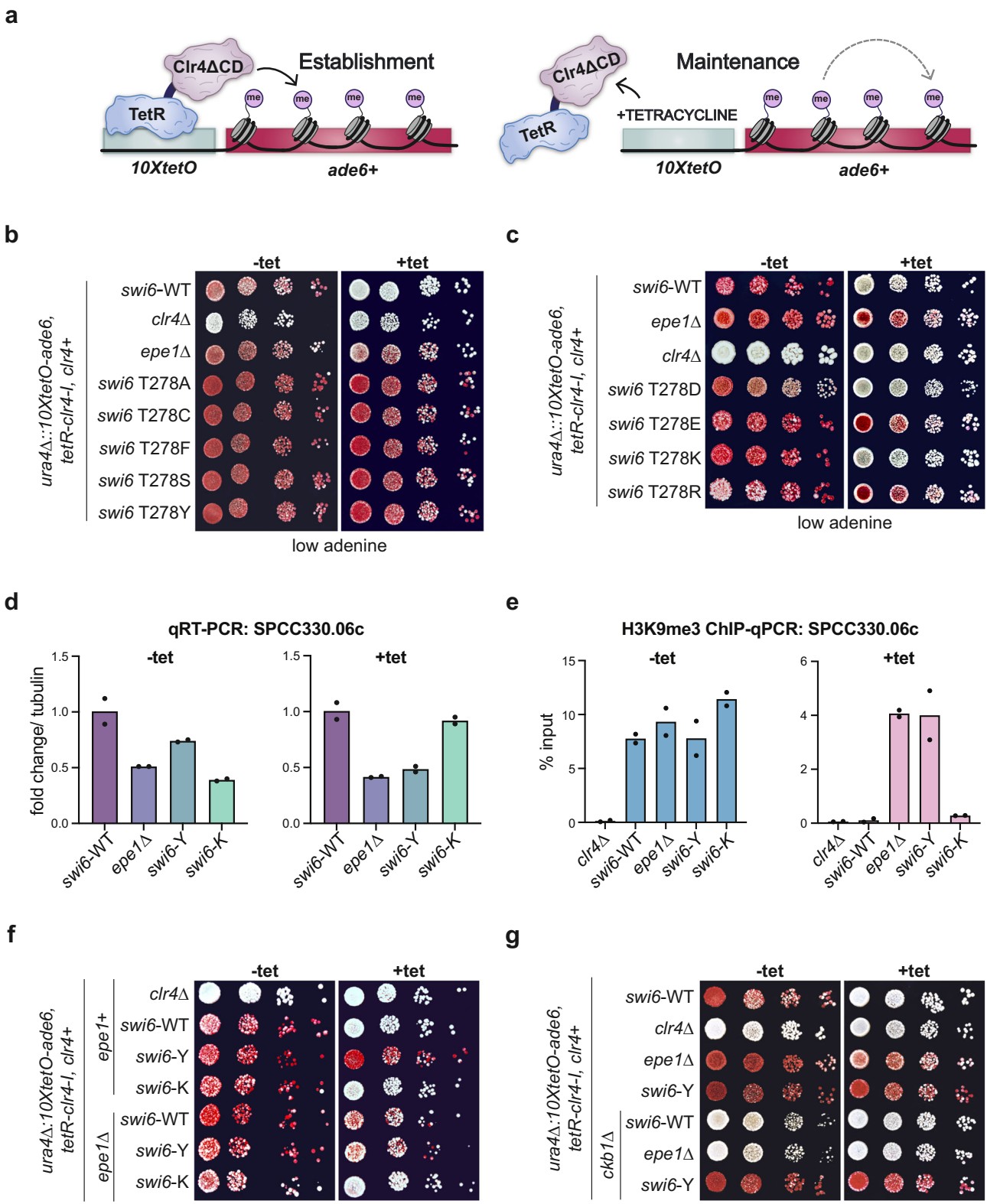

**Fig. 1 | Novel Swi6 variants affect heterochromatin maintenance. a** Schematic of TetR-Clr4-I recruitment to *10XtetO* binding sites upstream of an *ade6+* reporter gene, initiating heterochromatin establishment. Addition of tetracycline (+tet) releases TetR-Clr4-I, enabling measurements of heterochromatin maintenance uncoupled from sequence-specific establishment. **b**, **c** Silencing assay of *ura4Δ::10XtetO-ade6+* reporter in indicated genotypes in the absence (-tet) and presence (+tet) of tetracycline. Red cells indicate *ade6+* silencing. Cells are plated as 10-fold serial dilutions. **d** qRT-PCR measuring RNA levels at *SPCC330.06c* in indicated genotypes before (-tet) and after (+tet) tetracycline addition (N = 2). **e** ChIP-qPCR measuring H3K9me3 at *SPCC330.06c* in indicated genotypes before (-tet) and after (+tet) tetracycline addition (N = 2). **f, g** Silencing assay of *ura4Δ::10XtetO-ade6+* reporter in indicated genotypes in the absence (-tet) and presence (+tet) of tetracycline. Red cells indicate *ade6+* silencing. Cells are plated at 10-fold serial dilutions. Source data are provided as Source Data File.

*SPCC330.06c* RNA levels as observed in *swi6*-WT (Fig. 1d). Following 24 hours of growth in +tet, we still observed reduced transcript levels in *swi6*-Y consistent with heterochromatin maintenance. In contrast, *swi6*-K cells showed increased transcript levels consistent with a loss of heterochromatin maintenance (Fig. 1d). We measured H3K9me3 levels at *SPCC330.06c* using chromatin immunoprecipitation followed by qPCR (ChIP-qPCR). As expected, we observed high H3K9me3 enrichment in the context of establishment (-tet) in *swi6*-Y and *swi6*-K. In contrast, H3K9me3 persisted during maintenance (+tet) only in the case of *swi6*-Y, but not *swi6*-K (Fig. 1e).

As mentioned previously, heterochromatin maintenance in our reporter system is critically dependent on the absence of H3K9me eraser Epe1, with *epe1*Δ cells being the primary genetic context in which we observed red or sectored colonies on +tet media. To determine if the observed maintenance phenotypes in *swi6*-Y and *swi6*-K are dependent on Epe1, we deleted Epe1 in both genetic backgrounds. We expected that *swi6*-Y gain of maintenance would be unaffected or further enhanced by *epe1*Δ, whereas *swi6*-K would acquire a maintenance phenotype upon deleting *epe1*. Unexpectedly, maintenance was not restored in *swi6*–K *epe1*Δ cells, as indicated by the continued appearance of white colonies in +tet medium (Fig. 1f, Supplementary Fig. 1d). Additionally, *swi6*-Y *epe1*Δ cells exhibited a slightly weaker epigenetic maintenance phenotype with fewer sectored colonies than *swi6*-Y *epe1*+ cells. Nevertheless, deleting *epe1*Δ in *swi6*-Y did not completely disrupt maintenance, as we still observed red and sectored colonies consistent with successful maintenance when cells were plated on +tet media (Fig. 1f, Supplementary Fig. 1d). These findings indicate *swi6*-Y has a persistent gain of function maintenance phenotype, whereas *swi6*-K, unlike *swi6*-WT, has a persistent loss of function maintenance phenotype.

Swi6 phosphorylation may be affected by Thr 278 substitutions. Therefore, we tested the effect of deleting Ckb1, a subunit of the casein kinase II complex (CK2) in *S. pombe* that phosphorylates Swi6. Loss of Ckb1-mediated phosphorylation disrupts heterochromatin silencing by inhibiting recruitment of the histone deacetylase Clr3 while also promoting Epe1 occupancy at sites of heterochromatin formation[39]. Upon deleting the CK2 subunit, *ckb1* (*ckb1*Δ), *ade6*+ silencing is lost in both *swi6*-WT and *epe1*Δ cells. Surprisingly, heterochromatin silencing remains intact in *ckb1*Δ *swi6*-Y cells, suggesting that the Swi6-Y gain of maintenance is not regulated by CK2 phosphorylation (Fig. 1g).

## Maintenance in Swi6-Y depends on H3K9me

To determine if heterochromatin spreading and epigenetic inheritance is affected in *swi6*-Y and *swi6*-K, we performed chromatin immunoprecipitation followed by sequencing (ChIP-seq) of H3K9me2 and H3K9me3. We observed large H3K9me domains consistent with successful heterochromatin establishment proximal to the *10XtetO-ade6*+ reporter site across all Swi6 variants (Fig. 2a, b). Upon +tet treatment, H3K9me2 and H3K9me3 levels are maintained in *swi6*-Y, like what we observe in *epe1*Δ cells, and lost in *swi6*-K, like what we observe in *swi6*-WT (Fig. 2a, b). These results further confirm that the gain of maintenance phenotype we observed in *swi6*-Y is dependent on the inheritance of H3K9me.

To determine the extent to which both Swi6 variants affect constitutive heterochromatin, we replaced the endogenous *swi6*+ gene with *swi6*-Y and *swi6*-K in cells where an *ade6*+ reporter was inserted at the pericentromeric outer repeats (*otr1R(SphI)::ade6 +*) (Fig. 2c). Unlike *swi6*-WT cells that appeared uniformly red, we observed a small proportion of white, *ade6*+ expressing colonies in *swi6*-Y and *swi6*-K, suggesting a minor defect in pericentromeric reporter gene silencing. These minor silencing defects were mirrored in our qRT-PCR analysis of pericentromeric (*dg* and *dh*) and telomeric (*tlh1*) transcripts (Fig. 2d). Nevertheless, enrichment for H3K9me2 and H3K9me3 at pericentromeres were comparable in *swi6*-Y and *swi6*-K expressing cells (Fig. 2e, Supplementary Fig. 2a–d) while slightly decreased at

telomeres in *swi6*-Y and *swi6*-K compared to *swi6*-WT cells. H3K9me2 and H3K9me3 levels at the rDNA locus were elevated in s*wi6*-Y but not in *swi6*-K, compared to *swi6*-WT (Supplementary Fig. 2a–d). We also observed elevated H3K9me2 and H3K9me3 enrichment at facultative heterochromatin islands such as meiotic genes (*mei4* and *ssm4*) in *swi6*-K cells compared to *swi6*-WT (Supplementary Fig. 2e–f).

## Swi6-Y and Swi6-K variants disrupt a direct interaction with Epe1

As previously noted, deleting *epe1*Δ did not affect the maintenance phenotypes associated with Swi6-Y or Swi6-K. We reasoned that Thr 278 substitutions might disrupt a direct binding interaction between Epe1 and Swi6. We mapped the position of Thr 278 within the Swi6-CSD relative to the dimerization interface using an X-ray crystallography-based model (Fig. 3a)[18]. The dimerization interface, consisting of two helices, facilitates hydrophobic contacts between two Swi6-CSD monomers via Leu 315. Disrupting the dimer by introducing a charged amino acid substitution (L315E or L315D) leads to a loss of silencing in vivo and loss of H3K9me binding specificity in vitro[18,40]. Interestingly, Thr 278 lies within a beta-sheet outside the dimerization interface with its side chain being solvent-exposed. This observation led to a hypothesis that Thr 278 could be involved in tuning Swi6-dependent protein-protein interactions.

We generated strains expressing a C-terminal V5-tagged Epe1 (Epe1-V5) to detect the Swi6-Epe1 interaction using coimmunoprecipitation assays (CoIP) (Fig. 3b). As expected, Swi6-WT copurifies with Epe1-V5 from cell lysates, consistent with the two proteins directly interacting in vivo[37]. In contrast, Swi6 is not detected in Epe1-V5 purifications from cells expressing Swi6-Y or Swi6-K variants. This observation suggests the Swi6-Epe1 interaction is disrupted in cells expressing Swi6-Y and Swi6-K, irrespective of whether the mutants exhibit a gain of maintenance or a loss of maintenance phenotype (Fig. 1b, c). To further evaluate Epe1 binding, we performed a pulldown-based binding assay using recombinant FLAG-Swi6 and MBP-Epe1 (Supplementary Fig. 3a). Consistent with our CoIP results, we detected an interaction between Epe1 and FLAG-Swi6-WT but not in the case of FLAG-Swi6-K in vitro. In fact, FLAG-Swi6-K abolished Epe1 binding to a similar extent as FLAG-Swi6 L315E, a mutation known to completely disrupt Swi6 dimerization and all PxVxL-dependent protein interactions[23].

Consistent with our *ade6*+ establishment phenotypes and Epe1 binding observations, *swi6*-Y and *swi6*-K cells grown in -tet media did not exhibit any additional enrichment of H3K9me2 and H3K9me3 in *epe1*Δ cells compared to *epe1*+ cells (Figs. 1f, 3c, d). However, H3K9me2 and H3K9me3 enrichment is selectively observed only in *swi6*-Y *epe1*Δ but not *swi6*-K *epe1*Δ cells when grown in +tet media, which is consistent with their *ade6*+ maintenance phenotypes (Figs. 1f, 3c, d).

To determine the molecular basis for how Swi6-Y or Swi6-K affects Epe1 binding, we used AlphaFold2 Multimer (AF-M) to generate a structural model of the interaction between Epe1 and Swi6[41]. First, we generated a structural prediction of the Swi6-CSD dimer. All five models predicted the dimer as expected, with the highest-ranking model aligning to the published crystal structures with an RMSD value of 0.49 Å (Supplementary Figs. 3b and 4). Next, we generated a structural prediction of the Swi6-CSD dimer and Epe1ΔC (amino acids 1–600), which we previously showed is sufficient to strongly interact with Swi6 (Supplementary Fig. 3c)[37]. We generated five models, all confidently predicting Epe1ΔC binds to the Swi6 dimerization interface via a PxVxL-like IGVVI sequence (Supplementary Figs. 3c and 5, residues 569-573). Consistent with known HP1 interacting motifs, the critical feature of this interacting sequence is a central valine residue. Additionally, we observed a second interaction interface- namely, a predicted helix in Epe1 that interacts with the auxiliary Swi6 beta-sheet surface containing Thr 278. We deleted the regions of Epe1 predicted to interact with the CSD dimer (residues 566-600) to generate three alleles of Epe1 (*epe1*Δ566-600, *epe1*Δ569-573, and *epe1*Δ577-589).

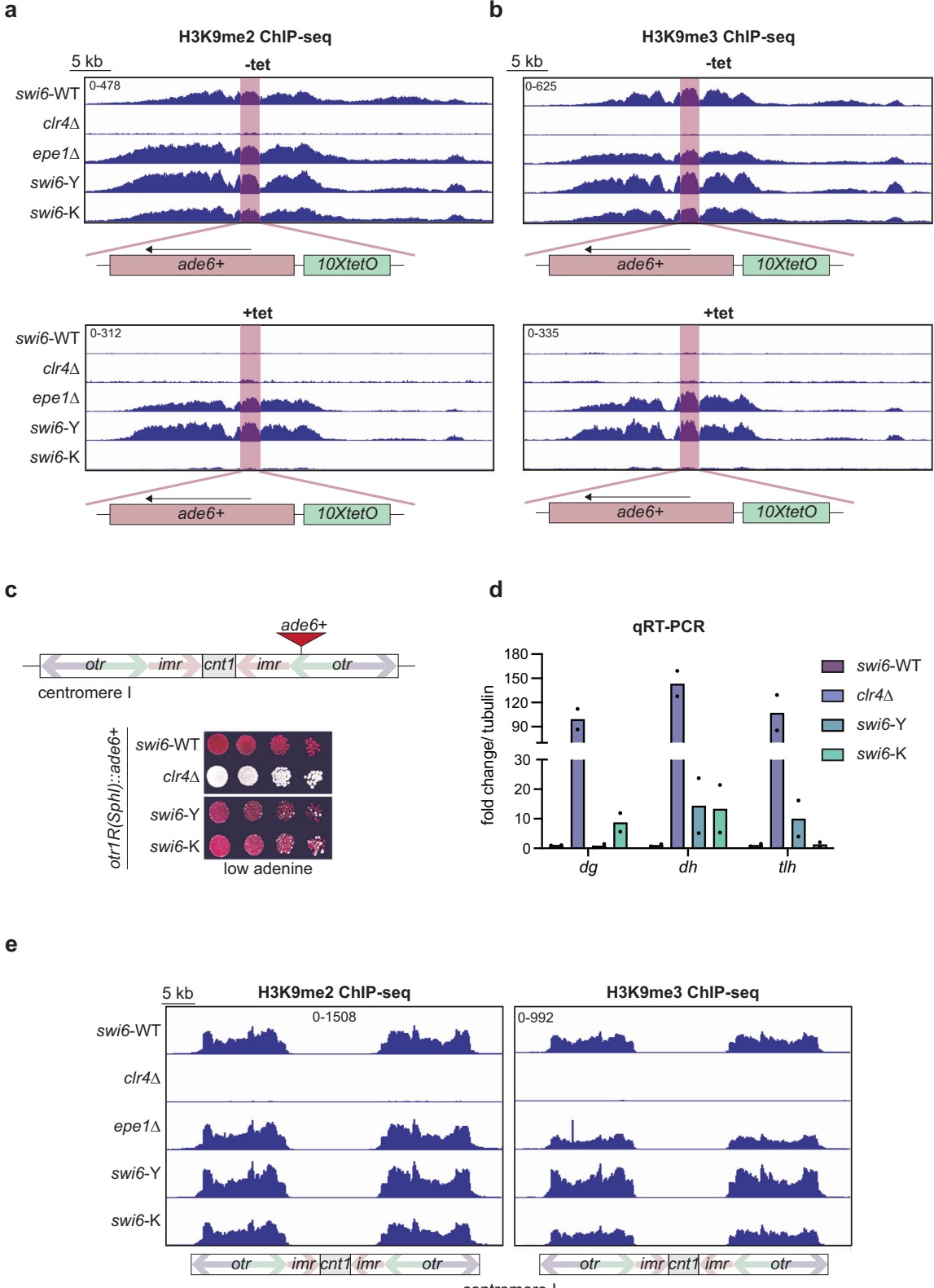

**Fig. 2 | Maintenance in Swi6-Y depends on H3K9me. a, b** ChIP-seq of H3K9me2 (**a**) and H3K9me3 (**b**) surrounding the *ura4Δ::10XtetO-ade6+* reporter in indicated genotypes and tetracycline treatment. The *ura4Δ::10XtetO-ade6+* reporter is highlighted in red. Each ChIP-seq track corresponds to a 40 kb region. Enrichment in all samples is shown as normalized reads per kilobase million (RPKM). **c** Top- Schematic detailing the *otr1R::ade6+* reporter, where *ade6+* is inserted within the outer pericentromeric repeats. Bottom- Silencing assay of *otr1R::(SphI)ade6+* reporter in indicated genotypes. **d** qRT-PCR measuring RNA levels at *dg*, *dh*, and *tlh* in indicated genotypes and tetracycline treatment (*N* = 2). **e** ChIP-seq of H3K9me2 and H3K9me3 at the centromere on chromosome 1 in indicated genotypes. Each ChIP-seq track corresponds to a 45 kb region with features within the centromere indicated in the schematic below. Enrichment in all samples is shown as normalized reads per kilobase million (RPKM). Source data are provided as Source Data File.

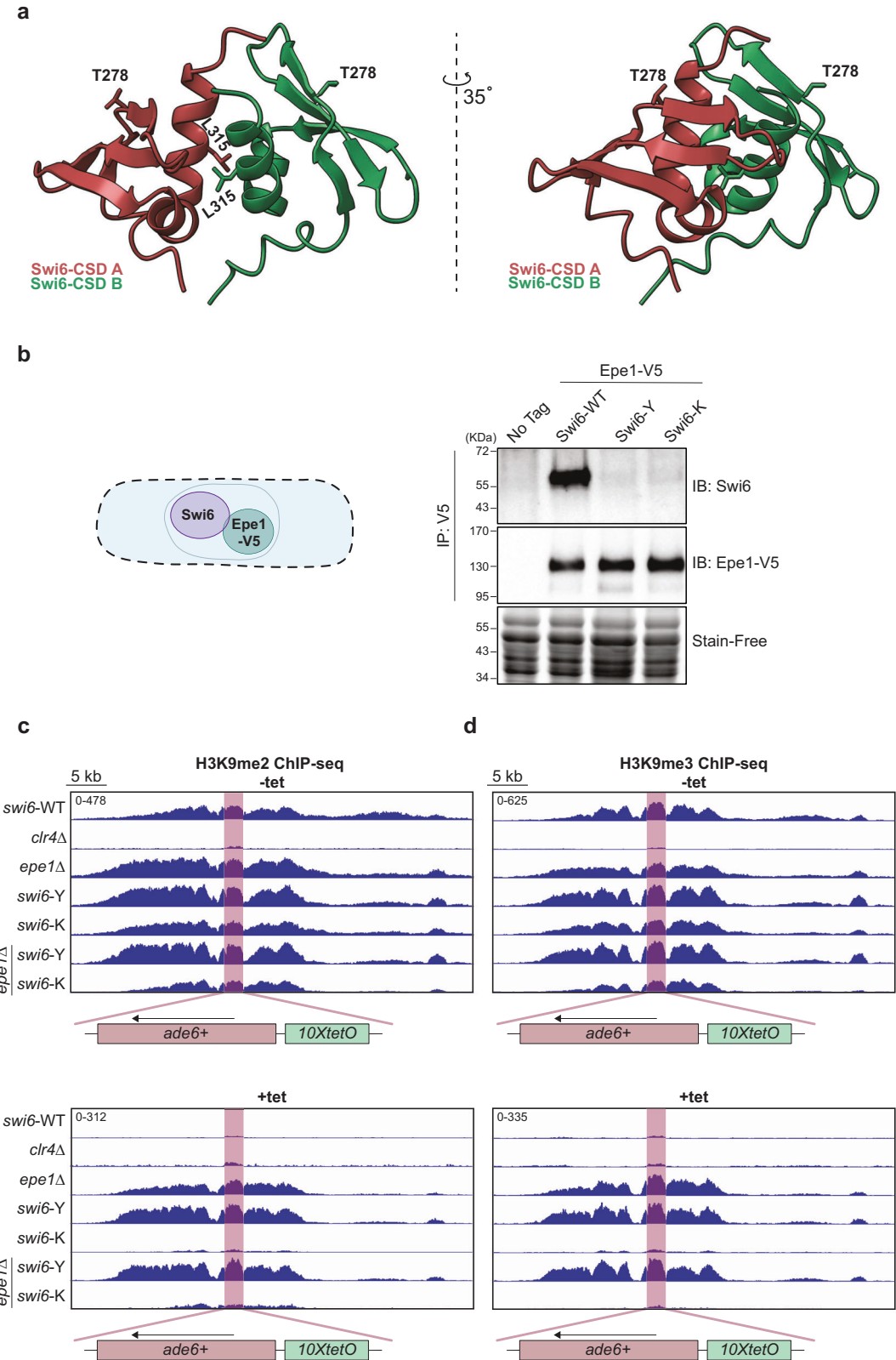

**Fig. 3 | Swi6-Y and Swi6-K variants disrupt a direct interaction with Epe1. a** X-ray crystallography structure of a Swi6-CSD dimer (PDB 1E0B, 1.90 Å). The dimerization interface and the side chains of Leu 315, a residue crucial for dimerization, are labeled. Thr 278 maps to a beta sheet interface with its side chain facing outward away from the dimerization interface. **b** Western blots of V5 coimmunoprecipitation (CoIP) performed with cell lysates to test the interaction between Epe1-V5 and Swi6. Epe1 is detected using a V5 antibody and Swi6 is detected using a primary antibody. **c**, **d** ChIP-seq of H3K9me2 (**c**) and H3K9me3 (**d**) surrounding the *ura4Δ::10XtetO-ade6+* reporter in indicated genotypes and tetracycline treatment. The *ura4Δ::10XtetO-ade6+* reporter is highlighted in red, displayed in a 40 kb region. Enrichment in all samples is shown as normalized reads per kilobase million (RPKM). Source data are provided as Source Data File.

Consistent with a loss of interaction between Swi6 and Epe1, we observe an increase in red or sectored colonies when cells were plated on +tet, consistent with heterochromatin maintenance resembling what we typically observed in *epe1Δ* cells (Supplementary Fig. 3d, *epe1Δ566-600*, *epe1Δ569-573*, and *epe1Δ577-589 compared to swi6-WT*). Additionally, we confirmed Epe1 protein levels are comparable between Epe1Δ566-600, Epe1Δ569-573, Epe1Δ577-589, and full-length Epe1-WT (Supplementary Fig. 3f). To determine if the interaction with the auxiliary surface where Thr 278 is embedded is generalizable, we generated a structural prediction of the Swi6-CSD dimer with a known interactor, Sgo1 (Supplementary Figs. 3e and 6). We observed similar interactions in the Sgo1-Swi6 predicted structure. A helix within Sgo1, distal to the PxVxL motif, interacts with the auxiliary Swi6 beta-sheet interface containing Thr 278. These predictions suggest that the beta-sheet within the Swi6 CSD serves as a binding site outside of the dimerization interface that could dictate Swi6 binding partner specificity.

### Swi6-Y and Swi6-K exhibit increased chromatin occupancy in vivo

We considered whether the molecular basis for the divergence in phenotypes between Swi6-Y and Swi6-K might arise from biochemical differences. We recombinantly expressed and purified different Swi6 variants (including Swi6-WT) from *E. coli* and analyzed their dimerization and nucleosome binding properties in vitro (Fig. 4a–c, Supplementary Figs. 7 and 8). We used mass photometry to measure the relative abundance of Swi6 species across a low nanomolar concentration range (2.5–20 nM). Mass Photometry (MP) is a single-molecule approach that uses light to detect the number and molar mass of unlabeled molecules in dilute samples and, given its measurement range, we expected to detect mass differences between Swi6 monomers and dimers. We detected two molecular species with the predicted masses for a Swi6 monomer (37 kDa) and a Swi6 dimer (74 kDa) in Swi6-WT, Swi6-Y, and Swi6-K (Supplementary Fig. 7a–e). The dimer population (74 kDa) was not detected in Swi6 L315E, a mutation that disrupts Swi6 dimerization (Supplementary Fig. 7c). The observed monomer-to-dimer ratios were consistent with concentration-dependent dimer formation. We predominantly observed dimers at our lowest measured concentration (2.5 nM), with monomers accounting for roughly 20% of the population (Supplementary Fig. 7d). We determined apparent dimerization constants using our observed relative abundance values ($K_{dim}$, Supplementary Fig. 7f)[42]. Consistent with previous work, Swi6-WT dimerizes with an apparent $K_{dim}$ of 0.38 nM. We did not observe a significant change in dimerization affinity in Swi6-Y and Swi6-K, with apparent $K_{dim}$ values being 0.27 nM and 0.20 nM, respectively[43]. Although there may be modest differences in dimerization that fall outside the detection limits of mass photometry, our results suggest dimerization is not significantly impacted by introducing Swi6-Y or Swi6-K mutations in vitro.

To measure Swi6-nucleosome binding affinity and specificity, we performed electrophoretic mobility shift assays (EMSAs) using reconstituted H3K9me0 or H3K9me3 mononucleosomes (Fig. 4a–c, Supplementary Fig. 8). We observed a shift of unbound nucleosomes to higher molecular weight species as Swi6 binds in a concentration-dependent manner. We determined that Swi6-WT binds to H3K9me3 mononucleosomes with an apparent $K_{1/2}$ of ~50 nM, which was very similar to the $K_{1/2}$ for Swi6-Y binding to H3K9me3 nucleosomes (~39 nM). In addition, both Swi6-WT and Swi6-Y bind to H3K9me3 mononucleosomes with similar specificity (2.4-fold for Swi6-WT and 2.3-fold for Swi6-Y) (Fig. 4a–c). As expected, we also observed a loss of specificity for H3K9me3 binding in Swi6 L315E (Fig. 4c). We detected no substantive differences between Swi6-WT and Swi6-Y in our in vitro nucleosome binding assays.

In vitro binding assays using mononucleosomes do not accurately reflect how Swi6 binds to chromatin in vivo, likely due to differences in substrate length and complexity[40,44]. To determine how Swi6-Y and Swi6-K bind to H3K9me in a native chromatin context, we mapped the dynamics of individual Swi6 molecules in living cells (Fig. 4d)[44]. We tracked PAmCherry-Swi6-Y and PAmCherry-Swi6-K dynamics and compared them to the dynamics of PAmCherry-Swi6-WT. For each mutant, we identified mobility states that best described the single-molecule trajectories we measured. Each state has a defined population and an average diffusion coefficient (Fig. 4e, f)[44,45]. Swi6-WT has four mobility states, namely a fast-state (unbound Swi6), two intermediate states (nucleic acid-associated Swi6, and unmethylated H3K9 chromatin-associated Swi6), and a slow-state (H3K9me2/3 chromatin-bound Swi6). Swi6-Y and Swi6-K exhibited only three mobility states in contrast to the four states we typically observed in the case of Swi6-WT (Fig. 4e, f). The most prominent change in our mobility state measurements was a 2-fold reduction in the fraction of Swi6 molecules across the two variants that occupy the fast-mobility state, corresponding to unbound Swi6. The fraction of molecules in the intermediate mobility state, corresponding to nucleic acid-associated Swi6, was at the limit of our analysis and hence not observed in either Swi6-Y or Swi6-K mutants. Most notably, the fraction of Swi6 molecules in the mobility states that correspond to H3K9me chromatin-binding increased, with an estimated shift from 50% of Swi6-WT molecules being chromatin-bound to ~90% of Swi6-Y and Swi6-K molecules being chromatin-bound. We additionally analyzed our data using a posterior distribution analysis (DPSP) to avoid overfitting biases that may arise from Bayesian methods[46]. The posterior distribution analysis with DPSP also revealed a greater proportion of molecules in low-mobility states (chromatin-bound) with a concomitant decrease in highly mobile (free) molecules in Swi6-Y and Swi6-K compared to Swi6-WT (Fig. 4g). The DPSP analysis supports our findings of an increase in the number of trajectories corresponding to low mobility molecules in Swi6-Y and Swi6-K compared to Swi6-WT. Given Swi6-Y and Swi6-K have divergent maintenance phenotypes, mechanisms other than increased chromatin occupancy must contribute to this process.

### The Swi6-rixosome interaction modulates epigenetic inheritance

Since the loss of Epe1 binding alone cannot explain the differential maintenance phenotypes we observed in Swi6-Y and Swi6-K, we investigated how all possible Swi6-dependent protein interactions are affected across the two variants using quantitative mass spectrometry[26]. We generated N-terminal FLAG-Swi6 strains to perform tandem mass tag affinity purification mass spectroscopy (TMT-AP-MS, Supplementary Fig. 9a). The full list of interacting proteins can be found in Supplementary Data 1. We compared protein interactions that were significantly altered in Swi6-Y and Swi6-K relative to Swi6-WT (Fig. 5a, b). In agreement with our CoIP measurements, Epe1 and its known binding partner Bdf2 are downregulated ~6-fold in Swi6-Y and Swi6-K (Fig. 5a–c)[47]. We also observed selective and significant upregulation of factors with known roles in ribosomal RNA (rRNA) processing, ribosome biogenesis, or nucleolar localization (~50% of all upregulated interactions) in the case of Swi6-Y but not Swi6-K. Among this group are the core subunits of the conserved rRNA processing and RNA degradation complex, the rixosome (Fig. 5a, c)[9,48]. We also noted a significant differential association of factors belonging to GO term categories, which include heterochromatin regulation, RNA polymerase II-mediated transcription, chromatin remodeling, RNA processing, DNA damage/cell cycle, and DNA replication (Fig. 5c, d). We did not observe significant changes in interactions with heterochromatin-associated factors that have known functions in epigenetic inheritance, which include subunits of the H3K9 methyltransferase CLRC complex, a deacetylase-remodeler complex, SHREC,

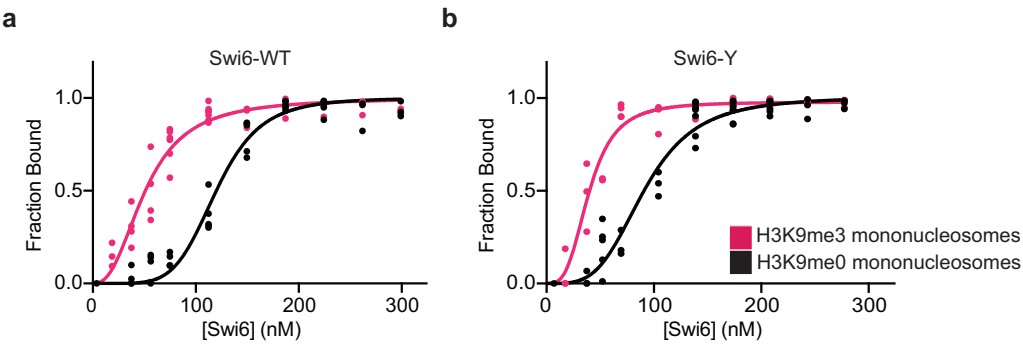

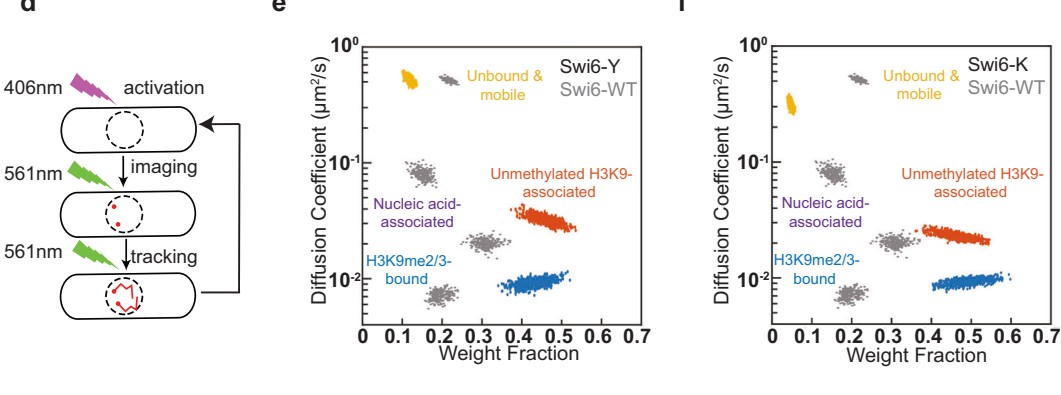

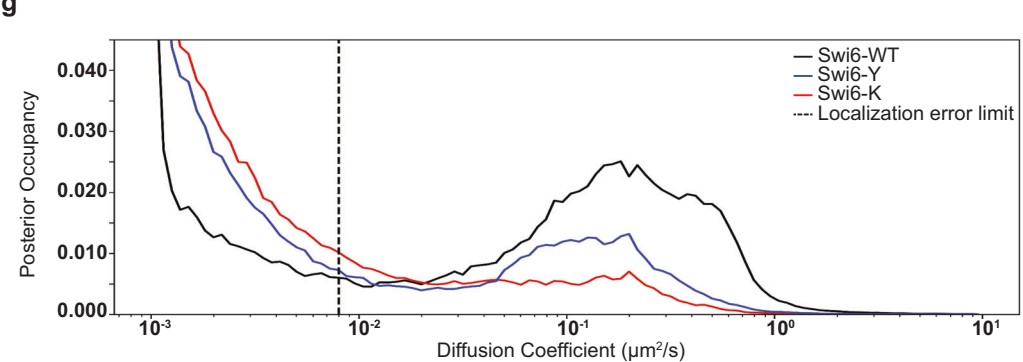

the histone chaperone complex. FACT, nuclear pore complex sub-units, and the nucleosome remodeler, Ino80 complex (Supplementary Fig. 9b)[9,26,49–51].

Our quantitative mass spectrometry results suggest the gain of maintenance phenotype we observed in Swi6-Y is determined by the extent of interaction with subunits of the rixosome complex. To test this hypothesis, we generated a separation of function mutation in the Grc3 subunit of the rixosome (*grc3* V70M). This mutation disrupts the rixosome-Swi6 interaction without affecting its ribosome biogenesis functions[48]. *swi6*-Y *grc3* V70M cells exhibited red colonies on -tet media, consistent with successful heterochromatin establishment. However, cells plated on +tet media turned white, consistent with a loss of *ade6+* silencing and selective disruption of maintenance (Fig. 5e). Therefore, the enhanced maintenance phenotype in Swi6-Y is critically dependent on its specific interaction with the rixosome complex.

**Fig. 4 | Swi6-Y and Swi6-K exhibit increased chromatin occupancy in vivo.**
**a**, **b** Concentration dependence curves of quantified electrophoretic mobility shift assays (EMSA) using H3K9me0 (black) and H3K9me3 (pink) mononucleosomes in (**a**) Swi6-WT and (**b**) Swi6-Y. Error bars indicate SD ($N = 5$). **c** Table summarizing the apparent binding metric ($K_{1/2}$) and specificity values observed for Swi6-WT, Swi6-Y and Swi6-L315E. **d** Schematic depicting single-molecule microscopy live-cell tracking workflow. PAmCherry-Swi6 molecules are photoactivated (406 nm) then imaged and tracked until photobleaching (561 nm, 25 frames/sec). The cycle is repeated 10–20 times/cell. **e**, **f** Mobility states detected by SMAUG analysis for (**e**) PAmCherry-Swi6-Y and (**f**) PAmCherry-Swi6-K. Each point is the average single-molecule diffusion coefficient of Swi6 following a single iteration of the Bayesian

algorithm after convergence. Mobility states are color-coded by fast diffusing unbound state (yellow), moderately diffusing nucleic acid-associated state (purple), slow diffusing unmethylated H3K9 bound state (red), and slow diffusing H3K9me2/3 bound state (blue). Dataset: 17150 single-molecule steps from 2039 PAmCherry-Swi6-Y trajectories and 67 cells. 19718 single-molecule steps from 1225 PAmCherry-Swi6-K trajectories and 68 cells. Mobility states determined in Swi6-WT (gray) are plotted as a reference. **g** Posterior occupancy across diffusion coefficients for PAmCherry-Swi6-WT (black), Swi6-Y (blue), and Swi6-K (red) with DPSP analysis[81]. The dashed line represents the localization error limit separating fast-diffusing molecules (right) from slow-diffusing molecules (left). Source data are provided as Source Data File.

We considered the possibility that a change in rixosome binding to Swi6-K may correspond only to the chromatin bound fraction. To test this, we expressed a Crb3-TAP fusion protein in cells followed by ChIP-qPCR using protein A binding to measure its occupancy at pericentromeric repeats (*dg*) and the *10XtetO-ade6+* reporter locus (*SPCC.330.06c*) (Fig. 5f). We detected a small, but significant decrease in rixosome binding at both loci (1.7-fold) in Swi6-K relative to Swi6-WT. Consistent with our mass spectrometry findings, we observe a significant increase in rixosome binding at pericentromeric repeats (3.9-fold) and the reporter gene locus (3-fold) in Swi6-Y. These results suggest the persistent loss of maintenance observed in Swi6-K is likely due to reduced rixosome binding at heterochromatin (Fig. 5f). These observations also explain why deleting Epe1 cannot rescue the maintenance defect in Swi6-K due to the persistent loss of rixosome interactions (Fig. 1f).

We generated a model of the Swi6-CSD dimer and Grc3 using AF-M (Supplementary Fig. 9c). Grc3 interacts with the Swi6-CSD dimer using a PxVxL-like motif, but the upstream and downstream contacts between Grc3 and the Swi6-CSD are strikingly different compared to what we observed in the case of Sgo1 and Epe1 (Supplementary Fig. 3c, f). The helix we previously noted in our Epe1-Swi6 and Sgo1-Swi6 structural models that interacts with the Swi6 auxiliary surface containing Thr 278 was notably absent in all five Grc3-Swi6 models (Supplementary Fig. 10). These in silico differences across Swi6 binding partners (Grc3, Sgo1, and Epe1) provide a potential molecular basis for tunability that depends on interactions with an auxiliary binding surface extending beyond the dimerization interface.

**Genetic rescue of Swi6-K heterochromatin maintenance defects**
Previous work shows that targeting the histone deacetylase Clr3 to heterochromatin is sufficient for epigenetic inheritance despite the presence of Epe1[37,49]. The HDAC activity of Clr3 reduces histone turnover, a characteristic feature of heterochromatin that is thought to promote epigenetic inheritance[52]. We tested if tethering Clr3 is sufficient to rescue defective maintenance in *swi6*-K expressing cells. We expressed a Gal4-Clr3 fusion protein in strains containing two orthogonal DNA binding sequences, i.e. *10XUAS* sites for Gal4 binding and *10XtetO* sites for TetR binding, both of which are placed upstream of the *ade6+* reporter gene (Supplementary Fig. 11a). Despite Epe1 being present, we observed robust maintenance of *ade6+* silencing, with cells appearing red or sectored when plated on +tet media (Supplementary Fig. 11a, *swi6*-WT, *gal4-clr3*). This process is critically dependent on Swi6 since both establishment and maintenance were eliminated in cells lacking Swi6 (Supplementary Fig. 11a, *swi6Δ*, *gal4-clr3*). Interestingly, tethering Clr3 rescued the Swi6-K maintenance defect since we observed both successful establishment (red colonies, -tet) and maintenance (red or sectored colonies in +tet), although maintenance was not nearly as robust as what we observed in *swi6*-WT cells (Supplementary Fig. 11a). Furthermore, targeting Clr3 could not bypass the requirement for the rixosome interaction in heterochromatin maintenance. In cells expressing *grc3*-V70M, tethering Clr3 failed to produce red or sectored colonies in cells plated on +tet media (Supplementary Fig. 11a). Hence, Clr3-mediated histone deacetylation

can compensate for defective heterochromatin maintenance in the case of *swi6*-K but not in the case of *grc3*-V70M.

We considered if compensatory mutations in Swi6 could restore protein interactions that are potentially lost in Swi6-K cells. To identify such mutations, we utilized the PCR-based targeted saturation mutagenesis approach described previously (Supplementary Fig. 1a) to generate a variant library at residues proximal to Swi6-K (residues: 268-277, 279-302). This screen revealed substitutions at Asp 283 to histidine (H), threonine (T), serine (S), arginine (R), and glutamic acid (E) that rescued the maintenance defect in the Swi6-K background, indicated by red or sectored colonies on +tet media (Supplementary Fig. 11b). This residue, D283, falls within a region of Swi6 involved in binding to the histone chaperone complex, FACT[51]. Collectively, our observations suggest the Swi6-K maintenance defect is likely also caused by changes in protein interactions also involving the rixosome but can be genetically restored through compensatory mutations that affect FACT or Clr3 binding.

## Discussion
HP1 proteins have a conserved architecture yet are functionally versatile. We hypothesized regions which exhibit high sequence variability within the otherwise conserved chromoshadow domain (CSD) contribute towards HP1 proteins acquiring functional properties. We show substitutions associated with a single amino-acid residue, Thr 278, within the Swi6-CSD influence the maintenance of an ectopic heterochromatin domain while preserving Swi6 dimerization, nucleosome binding, and transcriptional silencing. Our findings support a model where sequence variation outside the dimerization interface enables Swi6, and possibly other HP1 proteins, to reconfigure their functions. Hence, the plasticity of HP1 proteins arise not from changes to conserved sites but from alterations within auxiliary regions that preserve overall protein architecture. In silico modeling of Epe1, Sgo1, and Grc3, with the Swi6-CSD dimer show similarities in how PxVxL-like motifs bind to the Swi6-CSD dimer interface[23]. In contrast, our predictions show divergence in motifs that may interact with the Thr 278-containing beta-sheet interface (Supplementary Fig. 3c, e, and c). Although further validation is needed, our studies suggest the exist of an interface in HP1 proteins that can toggle effector binding specificity. Recent work on human HP1 alpha and its interacting partner INCENP underscores the importance of this second interaction site in tuning HP1-protein interactions demonstrating that the binding site we identified could be conserved in *S.pombe* and humans[53].

Structural studies to support our AF-M models are currently unavailable, however, NMR measurements have identified interactions between peptides of Clr3, Sgo1, and histone H2B with the Swi6-CSD dimer[22,43]. These data identified potential interactions involving Thr 278 and neighboring residues within Swi6. Hence, Thr 278 can function as a specificity determinant, enabling Swi6 to differentiate between different effectors, all of which share a PxVxL-like motif and bind to a common Swi6-CSD interface. Indeed, we have shown the Epe1-rixosome-Swi6 axis is sensitive to substitutions within the auxiliary beta sheet binding interface. Recent work has also shown that the Drosophila HP1 protein Rhino utilizes its chromodomain to interact

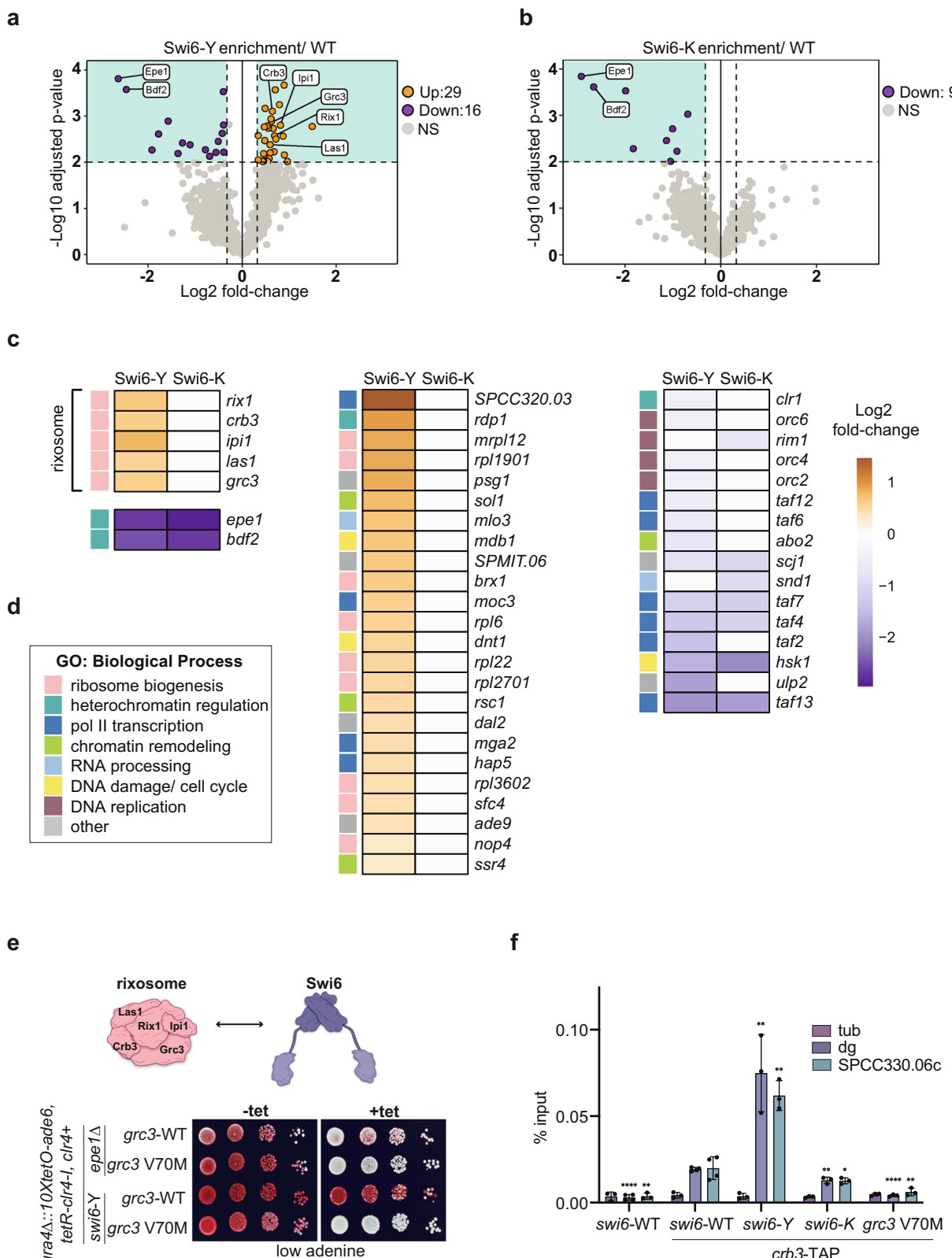

with the transcription factor Kipferl and mutations affecting Kipferl binding do not affect H3K9me recognition, suggesting there may be other potential interfaces that can be exploited to evolve different protein-protein interactions[54].

We aligned HP1 sequences across the *Schizosaccharomyces* lineage (*S.cryophilus*, *S.japonicus*, *S.octosporus*, and *S.osmophilus*) and found minimal sequence variation within the auxiliary beta-sheet

region across Swi6 orthologs (Supplementary Fig. 12). However, sequence conservation within this region significantly declines when comparing Swi6 to Chp2 or HP1 proteins from other organisms[23,30]. Hence, in addition to variations within PxVxL motif binding, we propose that this auxiliary beta sheet can further contribute to functional divergence between HP1 proteins[25]. Most substitutions, apart from the original Thr 278 residue, led to the persistent

**Fig. 5 | The Swi6-rixosome interaction modulates epigenetic inheritance.**
**a, b** Volcano plot displaying tandem mass tag mass spectrometry (TMT-MS) affinity purifications of (**a**) 3XFLAG-Swi6-Y and (**b**) 3XFLAG-Swi6-K. Samples were normalized to untagged Swi6 and the relative abundance over 3XFLAG-Swi6-WT were analyzed using a two-sided t-test. Shaded boxes mark a p-value = 0.01 (horizontal) and a 1.25-fold enrichment over the 3XFLAG-Swi6-WT reference (vertical) (N = 3), only interactions detected with > 1 peptide are included in the dataset. Proteins are color-coded as upregulated interactions (orange), downregulated interactions (purple), unenriched interactions (gray), and relevant heterochromatin regulators are labeled. The full dataset of interacting proteins can be found in Supplementary Data 1. **c** Heat map **c**omparing upregulated (orange) and downregulated (purple) interactions observed in Swi6-Y and Swi6-K, p-value cutoff is <0.01. **d** GO term analysis of biological process categories for upregulated and downregulated interactions. Each interaction within the (**c**) heat map is annotated with a colored box, denoting the corresponding GO-term biological process. The sample size for

each GO term is as follows: ribosome biogenesis N = 14, heterochromatin regulation N = 4, pol II transcription N = 10, chromatin remodeling N = 4, RNA processing N = 2, DNA damage/ cell cycle N = 3, DNA replication N = 4, other N = 6. **e** Top- Schematic of th**e** rixosome complex subunits. Bottom- Silencing assay of *ura4Δ::10XtetO-ade6+* reporter in indicated genotypes in the absence (-tet) and presence (+tet) of tetracycline. Red cells indicate *ade6+* silencing. Cells are plated at 10-fold serial dilutions. **f** ChIP-qPCR measuring Crb3-TAP at *tub*, *dg*, and *SPCC330.06c* in indicated genotypes. Error bars indicate SD (N = 3 or N = 4, replicates plotted for each sample). Mean of each sample were compared to their corresponding *swi6-WT* mean using an unpaired one-tailed t-test (P value cutoff <0.05) and the significance values are indicated as P < 0.05 (*), P < 0.01 (**), P < 0.0001 (****). The exact p-values are as follows for *dg* (*swi6*-WT -*crb3*-TAP < 0.0001, *swi6*-K = 0.0016, *swi6*-Y = 0.0018, and *grc3* V70M < 0.001) and *SPCC33.06c* (*swi6*-WT -*crb3*-TAP = 0.0013, *swi6*-K = 0.0256, *swi6*-Y = 0.0014, and *grc3* V70M = 0.005). Source data are provided as Source Data File.

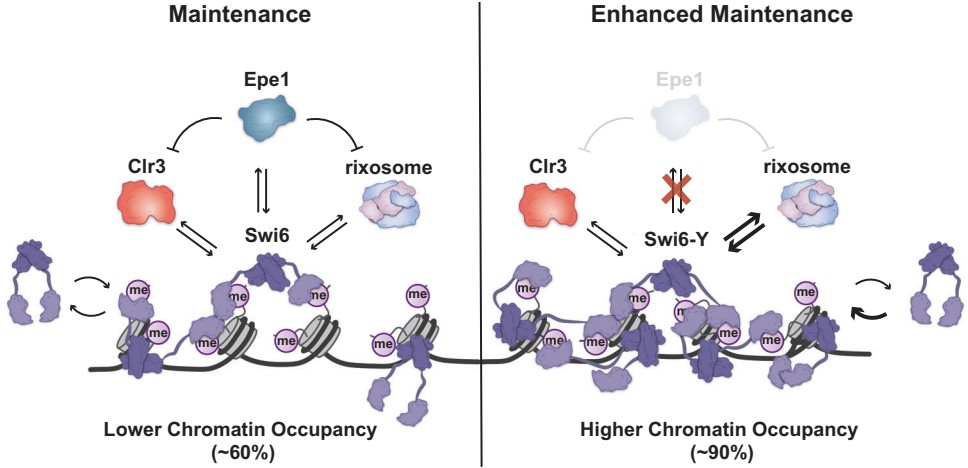

**Fig. 6 | Model for Swi6-CSD-mediated epigenetic inheritance capacity.** The CSD amino acid composition in Swi6-WT permits binding to effector proteins that inhibit (Epe1) and promote (Clr3 and rixosome) heterochromatin, leading to restriction of heterochromatin maintenance at the ectopic locus. Upon Swi6-Y mutation, the Epe1 interaction is lost and the rixosome interaction is upregulated, leading to enhanced heterochromatin maintenance. Upon Swi6-K mutation, the Epe1 interaction is lost but the rixosome interaction is not upregulated, leading to

defective heterochromatin maintenance. Swi6-Y and Swi6-K are more chromatin-bound than Swi6-WT, indicating the chromatin landscape is altered. The rixosome is recruited to heterochromatin directly by Swi6 for heterochromatin RNA clearance. Chromatin-associated transcripts are processed for subsequent degradation, allowing for Clr4-mediated read-write heterochromatin inheritance. Therefore, our findings support a model where Swi6-mediated effector recruitment, specifically Epe1 and the rixosome, influences epigenetic inheritance of heterochromatin.

gain of epigenetic inheritance, and a subset of charged amino acid substitutions led to a persistent loss of epigenetic inheritance (Fig. 1b, c). The consequences of both scenarios are absolute, with heterochromatin being inflexible and not regulatable. Therefore, our findings suggest Thr 278, and other proximal amino acid residues within the beta-sheet interface in Swi6 contribute to epigenetic plasticity wherein cells can invoke memory depending on changes in their physiology or environment. Notably, heterochromatin maintenance in *S. pombe* is responsive to environmental changes, which can be regulated by altering Epe1 availability in cells. Epe1 expression is sensitive to glucose availability, cAMP levels, stress, and ubiquitination[55–58]. Furthermore, post-translational modifications of Swi6, such as phosphorylation, can alter the balance of interactions between the histone deacetylase Clr3 and H3K9 demethylase Epe1[39].

The divergent phenotypes associated with Swi6-Y and Swi6-K present an interesting case study for how minimal changes within an HP1 protein, where protein architecture is preserved, can serve as drivers of functional innovation. It is evident from our biochemistry and mass spectrometry data that these divergent effects are not due to differences in protein structure (Swi6-Y and Swi6-K form stable dimers), nucleosome binding (site-specific perturbations do not affect H3K9me binding), or interactions with Epe1. Furthermore, both variants exhibit increased chromatin occupancy in vivo yet produce

opposite effects on epigenetic inheritance. Instead, the primary decision point for epigenetic inheritance in *S. pombe* is the extent to which the H3K9me bound Swi6 fraction efficiently interacts with components of the rixosome complex and recruits them to heterochromatin[9,48].

We have previously shown that Epe1 mutations, which are thought to affect catalytic activity, lead to a loss of a direct interaction between Epe1 and Swi6[37]. This trade-off in protein-protein interactions we observed in Swi6-Y lends additional support to a model where Epe1 regulates epigenetic inheritance by attenuating the interaction between Swi6 and heterochromatin maintenance enhancers, such as the rixosome (Fig. 6). In contrast, a reduction in the chromatin associated rixosome fraction leads to the observed loss of maintenance in Swi6-K. We can rescue the maintenance defect observed in Swi6-K two ways: through D283 compensatory mutations that can affect FACT complex binding or by tethering Clr3 at the ectopic locus which would lead to reduced histone turnover (Supplementary Fig. 11).

Our observations support a model where the basic unit of epigenetic inheritance involves multiprotein, Swi6-dependent complexes that assemble using H3K9me chromatin as a template[12]. This model is consistent both with Swi6-Y having increased chromatin occupancy and increased interactions with heterochromatin maintenance enhancers such as the rixosome. We envision that the stable

association of Swi6 before and after DNA replication could have a crucial role in ensuring that epigenetic states remain stable and heritable across multiple generations. Envisioning the potential protein-protein interaction network that emerges from Swi6 binding to heterochromatin makes its inheritance unique and distinct from euchromatin.

## Methods

### Strains

All mutation, deletion, and tagging strains were generated using published standard protocols, which include PCR-based gene targeting, the SpEDIT CRISPR/Cas9 system, or by a cross followed by random spore analysis[59,60]. All strains used in this study are listed in Supplementary Data 2 and oligos are listed in Supplementary Data 3.

### Site-directed saturation mutagenesis library generation

The site-directed saturation mutagenesis is adapted from ref. [61]. Primers were designed to contain an "NNN" degenerate sequence at every codon position within the Swi6 coding sequence and were commercially synthesized. The mutagenesis primer sequences are listed in Supplementary Data 3. A Swi6 plasmid template was generated using topoisomerase-based cloning. The insert sequence was generated by PCR from genomic DNA and included the Swi6 coding sequence with 500-bp flanking homology segments. The library was generated by three-step PCR. The first PCR reaction introduces the degenerate codons, the second extends the truncated PCR products from step 1, and the third adds the necessary homology for recombination at the endogenous Swi6 locus. The library was transformed into a strain where the endogenous Swi6-CSD is deleted with a *ura4-kan* selection marker. Positive transformants were selected by growth on FOA and loss of G418 resistance. Red or pink colonies on YE (establishment) were subsequently tested on YE+tet (maintenance). Colonies that show maintenance were considered library hits, and the Swi6 mutation was mapped using Sanger sequencing.

### Expression and purification of recombinant protein

Swi6 expression and purification approach was adapted based on an earlier study[44]. Swi6 coding sequence was cloned into N-terminal 6XHis-tag–containing pET vectors, and mutants were generated using site-directed mutagenesis. All Swi6 proteins were purified from BL21(DE3)-RIPL *E. coli* cells. Cells were grown at 37 °C to OD 0.5 to 0.8 in LB media with ampicillin (100 µg/ml), induced with 0.4 mM isopropyl-β-D-thiogalactopyranoside (IPTG), and were grown for 16 hours at 18 °C. Cells were harvested and resuspended in lysis buffer [1× phosphate-buffered saline (PBS) buffer (pH 7.3), 300 mM NaCl, 10% glycerol, 0.1% Igepal CA-630, 1 mM phenylmethylsulfonyl fluoride (PMSF), aprotonin (1 µg/ml), pepstatin A, and leupeptin] and sonicated. Cell debris was removed by centrifugation at 25,000 x *g* for 35 min. Cell lysates were incubated with HisPur NiNTA resin (Thermo Fisher Scientific) at 4 °C for at least 2 hours. The resin was washed with lysis buffer, and protein was eluted [20 mM Hepes (pH 7.5), 100 mM KCl, 10% glycerol, and 500 mM imidazole] and incubated with the corresponding protease (Ulp1 or TEV) overnight at 4 °C. After cleavage of 6XHis-tag, the products were further isolated by anion exchange chromatography using a HiTRAP Q HP column (Cytiva) and size exclusion chromatography using a Superdex200 10/300 (Cytiva) column. Proteins were dialyzed into storage buffer [20 mM Hepes, 100 mM KCl, 10% glycerol, and 1 mM dithiothreitol (DTT)]. Protein concentrations were determined using ultraviolet (UV) absorption measurements at 280 nm and molecular weights (MWs) and extinction coefficients computationally determined for Swi6-WT (MW = 37,292.6 g/mol, $\varepsilon$ = 41,035 $M^{-1}$ $cm^{-1}$), Swi6-L315E (MW = 37,308.6 g/mol, $\varepsilon$ = 41,035 $M^{-1}$ $cm^{-1}$), Swi6-T278Y (MW = 37,354.7 g/mol, $\varepsilon$ = 42,525 $M^{-1}$ $cm^{-1}$), and Swi6-T278K (MW = 37,319.7 g/mol, $\varepsilon$ = 41,035 $M^{-1}$ $cm^{-1}$) using the Expasy ProtParam tool. Protein was further equalized using SDS-PAGE densitometry quantification. Epe1

was purified as described previously[37]. MBP-His-TEV-Epe1 was cloned into a pFastBac vector (Thermo Fisher Scientific) and used for Bacmid generation. Low-titer baculoviruses were produced by transfecting Bacmid into Sf21 cells using Cellfectin II reagent (Gibco). Full-length *S. pombe* Epe1 protein (wild-type and mutant) was expressed in Hi5 cells infected by high titer baculovirus, amplified from Sf21 cells. After 44 h of infection, Hi5 cells were harvested and lysed in buffer A (30 mM Tris-HCl (pH 8.0), 500 mM NaCl, 5 mM EDTA, 5 mM β-mercaptoethanol with protease inhibitor cocktails) using Emulsiflex-C3 (Avestin). The cleared cell lysate was applied to Amylose resin (New England Biolabs), followed by washing with buffer A and elution with buffer A containing 10 mM maltose. Proteins were further purified using a Superdex 200 (GE Healthcare) size exclusion column. The protein was concentrated in a storage buffer containing 30 mM Tris-HCl (pH 8.0), 500 mM NaCl, 30% glycerol, and 1 mM TCEP.

### Total protein extraction from *S. pombe*

To detect protein expression in strains containing Swi6 mutants and tagged proteins, protein extracts were prepared using Trichloroacetic Acid (TCA) precipitation. Strains were grown in liquid yeast extract supplemented with adenine (YEA) for 16 hours at 32 °C, and 7 ODs worth of cells were harvested. Cell pellets were washed with 1 mL of ice-cold water and resuspended in 150 µL of YEX Buffer (1.85 M NaOH, 7.5% beta-mercaptoethanol). After 10 minutes of incubation on ice, protein precipitation was performed by adding 150 µL of 50% TCA (~ 3 N) and mixing by inversion. The extracts were then incubated for 10 minutes on ice, pelleted by centrifugation, and excess TCA was carefully removed. The protein extracts were resuspended in 2X SDS sample buffer (125 mM Tris-Base pH 6.8, 8 M urea, 5% SDS, 20% glycerol, 5% BME) and centrifuged for 5 minutes to clear cell debris. Samples were analyzed by western blotting, as described below.

### Western blotting

Protein samples were resolved by gel electrophoresis on 4−20% Mini-PROTEAN® TGX Stain-Free™ Protein Gels. Immunoblotting was performed using the BioRad Trans-Blot Turbo Transfer System, and transfer was performed at 2.5 A and 25 V for 7 minutes onto 0.2 µm Nitrocellulose. Membranes were blocked using 5% non-fat dry milk in Tris-buffered saline pH 7.5 with 0.1% Tween-20 (TBST) for 1 hour. The membrane was then incubated with the primary antibody at an optimized concentration overnight at 4 °C. Following incubation, the membrane was washed 3 times with TBST and incubated with the appropriate secondary antibody (1:10,000 dilution) for 1 hour at room temperature. After the incubation, the membrane was washed 3 times with TBST and incubated with SuperSignal West Pico/Femto PLUS Chemiluminescent Substrate. The membrane was imaged for chemiluminescence on a Bio-Rad ChemiDoc. For quantitative Swi6 western, an Alexa Fluor 647 conjugated secondary antibody (A-21246, Invitrogen) was used (1:10,000 dilution) and imaged using the Alexa Fluor 647 filter on a BioRad ChemiDoc.

### Coimmunoprecipitation to detect Epe1-Swi6 interaction

CoIP experiments were performed as described previously[37]. 1.5 L of fission yeast cells were grown in YEA medium at 32 °C to an $OD_{600}$ = 3.5 and harvested by centrifugation. The cell pellets were washed with 10 ml TBS pH 7.5, resuspended in 1.5 ml lysis buffer (30 mM HEPES pH 7.5, 100 mM NaCl, 0.25% Triton X-100, 5 mM $MgCl_2$, 1 mM DTT), and the cell suspension was snap-frozen into liquid nitrogen drop-wise and cryogenically ground using a SPEX 6875D Freezer Mill. The frozen cell powder was stored at −80 °C, thawed at room temperature, and resuspended in an additional 10 ml of lysis buffer with a protease inhibitor cocktail and 1 mM PMSF. Cell lysates were subjected to two rounds of centrifugation at 39,000 x *g* for 5 and 30 mins. Protein levels were normalized for coimmunoprecipitation and immunoblot analysis using a Bradford Assay. Protein G Magnetic Beads were pre-incubated

with V5 antibody (A01724, Genscript) for 4 h and crosslinked with 10 volumes of crosslinking buffer containing 20 mM DMP (3 mg DMP/ml of 0.2 M Boric Acid pH 9) for 30 min at room temperature by rotating. Crosslinking was quenched by washing twice and incubated with 0.2 M ethanolamine pH 8 for 2 h at room temperature. The cell lysates were then incubated with antibody crosslinked beads for 3 h at 4 °C. Beads were washed thrice in 1 ml lysis buffer for 5 mins each, then eluted with 500 µl of 10 mM ammonium hydroxide. The ammonium hydroxide was evaporated using a speed vac (SPC-100H) for 5 h and resuspended in SDS sample buffer. Samples were resolved using SDS–polyacrylamide gel electrophoresis (SDS-PAGE) and transferred to PVDF membranes. Immunoblotting was performed by blocking the PVDF membrane in Tris-buffered saline (TBS) pH 7.5 with 0.1% Tween-20 (TBST) containing 5% non-fat dry milk and subsequently probed with desired primary antibodies and secondary antibodies. Blots were developed by enhanced chemiluminescence (ECL) method and detected using a Bio-Rad ChemiDoc Imaging System.

### In vitro binding assay to detect Epe1-Swi6 interaction

In vitro binding assays were performed by immobilizing recombinant 3X FLAG-Swi6 on 25 µl of FLAG M2 beads, which were incubated with three different concentrations of recombinant MBP-Epe1 fusion proteins in 600 µl binding buffer containing 20 mM HEPES pH 7.5, 150 mM NaCl, 5 mM MgCl$_2$, 10% glycerol, 0.25% Triton -X 100, 1 mM DTT. Reactions were incubated at 4 °C for 2 h and washed three times in 1 ml washing buffer (20 mM HEPES pH 7.5, 150 mM NaCl, 5 mM MgCl$_2$, 10% glycerol, 0.25% Triton -X 100, 1 mM DTT) for 5 min each, then 30 µl of SDS sample buffer was added followed by incubation at 95 °C for 5 min. Proteins were separated through SDS-PAGE and transferred to a PVDF membrane followed by incubation (1:10,000 dilution) with anti-MBP monoclonal antibody (E8032S, NEB) and M2 Flag antibody (A8592, Sigma). Western blot data for in vitro binding assays were analyzed using ImageJ software. The exposure times for the interaction assays were chosen and differed in each experiment to capture differences in the interaction between Epe1 and Swi6 depending on the assay conditions. Assays performed on different blots cannot be compared, but samples loaded on the same blot can be readily compared.

### Tandem-mass tag affinity purification mass spec

Protein levels were normalized between three technical replicates by silver stain. Dried eluates were sent to the Thermo Fisher Center for Multiplexed Proteomics at Harvard Medical School for further processing and analysis. Dried samples were resuspended in 20 mM EPPS, pH 8.5. Samples were reduced with TCEP, alkylated with iodoacetamide, and further reduced with DTT. Proteins were extracted with SP3 beads. Samples were digested overnight at room temperature with Lys-C, followed by digestion with trypsin for 6 hours at 37 °C. Protein samples were labeled with TMTPro reagents, and complete labeling was confirmed. All samples were pooled and desalted by stage-tip. Peptides were analyzed on an Orbitrap Eclipse Mass Spectrometer. MS2 spectra were searched using the COMET algorithm against an *S. pombe* Uniprot composite database (downloaded in 2023) containing its reversed complement and known contaminants. For proteome, Peptide spectral matches were filtered to a 1% false discovery rate (FDR) using the target-decoy strategy combined with linear discriminant analysis. The proteins were filtered to a < 1% FDR and quantified only from peptides with a summed SN threshold of >120. The following R packages were used for data analysis: tidyverse, ggplot RColorBrewer, SummarizedExperiment, matrixStats and pheatmap.

### Silencing assays

Cells were grown in 3 ml of yeast extract containing adenine (YEA) at 32 °C overnight. Cells were washed twice in water and then resuspended to a concentration of ~10$^7$ cells/ml. Ten-fold serial dilutions

(~5 µL) were plated on YE plates to evaluate establishment and YE + AHT to evaluate maintenance. Plates were incubated for 2–3 days before the results were cataloged.

### Chromatin immunoprecipitation (ChIP)

For H3K9me2 and H3K9me3 ChIP, 30 ml of cells were grown to late log phase (OD$_{600}$~1.8-2.2) in yeast extract supplemented with adenine (YEA) or YEA containing tetracycline (2.5 µg/ml) and fixed with 1% formaldehyde for 15 min at room temperature (RT). For TAP ChIP, 25 ml of cells were grown to late log phase (OD$_{600}$~1.8-2.2) in yeast extract supplemented with adenine (YEA) or YEA containing tetracycline (2.5 µg/ml) and then grown for 2 hrs at 18 °C. Cells were crosslinked with 1.5 mM ethylene glycol bis(succinimidyl succinate) (EGS) for 30 min at RT and then 1% formaldehyde for 30 min. 130 mM glycine was added to quench the reaction and incubated for 5 min at RT. The cells were harvested by centrifugation and washed with TBS (50 mM Tris, pH 7.6, 500 mM NaCl). Cell pellets were resuspended in 300 µL lysis buffer (50 mM HEPES-KOH, pH 7.5, 100 mM NaCl, 1 mM EDTA, 1% Triton X-100, 0.1% SDS, and protease inhibitors) to which 500 µL 0.5 mm glass beads were added. Cell lysis was carried out by bead beating using Omni Bead Ruptor at 3000 rpm x 30 sec x 10 cycles. Tubes were punctured, and the flow-through was collected in a new tube by centrifugation, which was subjected to sonication to obtain fragment sizes of roughly 100-500 bp long. After sonication, the extract was centrifuged for 15 min at 16,500 x *g* at 4 °C. The soluble chromatin was transferred to a fresh tube and normalized for protein concentration by Bradford assay. For each normalized sample, 25 µL lysate was saved as input, to which 225 µL of 1xTE/1% SDS were added (TE: 50 mM Tris pH 8.0, 1 mM EDTA). Protein A Dynabeads were pre-incubated with H3K9me2 antibody (Abcam, ab1220) or H3K9me3 antibody (39161, Active Motif). For each immunoprecipitation, we used 2 µg H3K9me2 antibody (Abcam, ab1220) and 2 µg H3K9me3 antibody (39161, Active Motif). For TAP ChIP, Dynabeads Pan Mouse IgG (Invitrogen, 11042) were used. Samples were incubated for 3 h at 4 °C. The beads were collected on magnetic stands and washed 3 times with 1 mL lysis buffer and once with 1 mL TE. For eluting bound chromatin, 100 µL elution buffer I (50 mM Tris pH 8.0, 10 mM EDTA, 1% SDS) was added, and the samples were incubated at 65 °C for 5 min. The eluate was collected and incubated with 150 µL 1xTE/0.67% SDS in the same way. Input and immunoprecipitated samples were incubated overnight at 65 °C to reverse crosslink. 60 µg glycogen, 100 µg proteinase K (Roche), 44 µL of 5 M LiCl, and 250 µL of 1xTE were added to each sample, and incubation was continued at 55 °C for 1 h. Phenol/chloroform extraction was carried out for all the samples, followed by ethanol precipitation. Immuno-precipitated DNA was resuspended in 100 µL of 10 mM Tris pH 7.5 and 50 mM NaCl. ChIP experiments were analyzed using quantitative PCR with Taq polymerase and SYBR Green using a CFX Opus 384 Real-Time PCR System. PCR primers are listed in Supplementary Table 1.

### ChIP-Seq library preparation and processing

ChIP-seq libraries were prepared and processed as described previously[13]. Libraries were constructed using the manufacturer's guidelines in the NEBNext® Ultra™ II FS DNA Library Prep Kit for Illumina, using 1 ng of starting material. Barcoded libraries were pooled and sequenced with next-generation sequencing. First, raw reads were demultiplexed by barcode. Then, the sequences were trimmed with Trimmomatic, aligned with BWA, and normalized by reads per million[62,63]. The reads were visualized with IGV 2.16.12.

### RNA extraction

10 mL of cells were grown to late log phase (OD$_{600}$~1.8-2.2) in yeast extract supplemented with adenine. Cells were resuspended in 750 µL TES buffer (0.01 M Tris pH7.5, 0.01 M EDTA, 0.5% SDS). Immediately, 750 µL of acidic phenol-chloroform was added and vortexed for

2 minutes. Samples were incubated at 65 °C for 40 minutes while vortexing for 20 seconds every ten minutes. The aqueous phase was separated by centrifuging in Phase Lock tubes for 5 minutes at 16,500 x g at 4 °C. The aqueous phase was transferred to new tubes, and ethanol precipitated. After extraction, RNA was treated with DNase. Then, the RNA was cleaned using RNeasy Mini kits (Qiagen). cDNA was prepared using oligodT and SuperScript III Reverse Transcriptase (Invitrogen). The cDNA was then used for qPCR with SYBR Green and Taq polymerase on a CFX OPUS 384 Real-Time PCR System. RNA levels were quantified using $\Delta C_T$ compared to tubulin (tub1) RNA levels. PCR primers are listed in Supplementary Data 3.

## Mass photometry

All mass photometry experiments were performed using full-length recombinant Swi6 protein at the Center for Macromolecular Interactions at Harvard Medical School using a Refeyn TwoMP instrument. Before taking measurements, the instrument was calibrated using a protein standard containing 10 nM β-amylase (Sigma Aldrich A8781) and 3 nM Thyroglobulin (Sigma-Aldrich 609310). Swi6 proteins were diluted to 100 nM in 20 mM HEPES pH 7.5, 100 mM KCl immediately before taking measurements. For each measurement, the objective was focused using 20 mM HEPES pH 7.5, 100 mM KCl, and the corresponding volume of 100 nM Swi6 was added to the droplet to achieve the desired final concentration (2.5-20 nM). Sample data was collected immediately. Figures and Gaussian fits of the resulting data were generated using the Refeyn DiscoverMP software. Apparent dimerization constants ($K_{dim}$) were determined by relative molecular abundance of the monomer and dimer populations at known Swi6 concentrations[42].

## AlphaFold2-multimer (AF-M) structural prediction

AlphaFold2 Multimer was used to predict protein-protein interactions using the Cosmic[2] Science Gateway server[41,64]. In all cases, we obtained 5 models with 3 recycles, and all structures were unrelaxed. Protein structures were plotted using ChimeraX-1.6.1[65]. We further analyzed our structures using a published pipeline to determine interface statistics from predicted multimer structural models[66]. This pipeline identifies all interchain interactions within 8 angstroms across all five models. The pipeline also provides several metrics to score the confidence of the predicted multimer interfaces accounting for the consistency of interactions and pLDDT scores across all models (Supplementary Figs. 4f, 5f, 6f, 10f).

## S. pombe live-cell imaging

Yeast strains containing a copy of PAmCherry-Swi6 or a PAmCherry-Swi6 mutant under the control of the native Swi6 promoter were grown in standard complete YES media (US Biological, catalog no. Y2060) containing the full complement of yeast amino acids and incubated overnight at 32 °C. This initial culture was diluted and incubated at 25 °C with shaking to reach an optical density at 600 nm (OD$_{600}$) of ~0.5. To maintain cells in an exponential phase and eliminate extranuclear vacuole formation, the culture was maintained at OD$_{600}$ ~0.5 for 2 days, diluting at ~12-hour intervals. To prepare agarose pads for imaging, cells were pipetted onto a pad of 2% agarose prepared in YES media with 0.1 mM N-propyl gallate (Sigma-Aldrich, catalog no. P-3130) and 1% gelatin (Millipore, catalog no. 04055) as additives to reduce phototoxicity during imaging. S. pombe cells were imaged at room temperature with a 100×1.40 numerical aperture (NA) oil-immersion objective in an Olympus IX-71 inverted microscope. First, the fluorescent background was decreased by exposure to 488 nm light (Coherent Sapphire, 200 W/cm2 for 20 to 40 s). A 406-nm laser (Coherent Cube, 405-100; 102 W/cm2) was used for photoactivation (200-ms activation time), and a 561-nm laser (Coherent Sapphire, 561-50; 163 W/cm2) was used for imaging. Images were acquired at 40-ms exposure time per frame. The fluorescence emission was filtered with a Semrock LL02-561-12.5 long-pass filter and a Chroma ZT488/561rpc 488/561 dichroic mirror to eliminate the 561 nm excitation source and imaged using a 512 × 512-pixel Photometrics Evolve EMCCD camera.

## Single-molecule trajectory analysis

Recorded Swi6-PAmCherry single-molecule positions were detected and localized with two-dimensional Gaussian fitting with home-built MATLAB software as previously described and connected into trajectories using the Hungarian algorithm[67–69]. These single-molecule trajectory datasets were analyzed by a nonparametric Bayesian framework to reveal heterogeneous dynamics[45]. This SMAUG algorithm uses nonparametric Bayesian statistics and Gibbs sampling to identify the number of distinct mobility states in the single molecule tracking dataset in an iterative manner. It also infers parameters, including weight fraction and average apparent diffusion coefficient for each mobility state, assuming a Brownian motion model. To ensure that even rare events were captured, we collected more than 10,000 steps in our single-molecule tracking dataset for each measured strain, and we ran the algorithm over >10,000 iterations to achieve a thoroughly mixed state space. The state number and associated parameters were updated in each iteration of the SMAUG algorithm and saved after convergence. The final estimation shows the data after convergence for iterations with the most frequent state number. Each mobility state is assigned a distinct color, and for each saved iteration, the average diffusion coefficient of that state is plotted against the weight fraction. The distributions of estimates over the iterations give the uncertainty in the determination of the parameters. For measurement of static molecules in fixed S. pombe cells, SMAUG converges to a single state with $D_{avg} = 0.0041 \pm 0.0003$ μm2/s. The average localization error for single-molecule localizations in this fixed-cell imaging is 32.6 nm. To benchmark against a model that overcomes the potential overfitting of the Bayesian model, we also applied the DPSP package to acquire the posterior occupancy distribution for diffusion coefficients of Swi6 and variants[46]. The DPSP package uses a Dirichlet process mixture model to acquire the posterior probability distribution for each dataset. The same trajectory datasets used in SMAUG analysis were stored in csv format and analyzed with the Python package DPSP under default parameter settings and corresponding pixel size and frame interval (https://github.com/alecheckert/dpsp).

## Nucleosome electrophoretic mobility shift assays (EMSAs)

Samples were prepared by varying concentrations of Swi6 while keeping substrate concentration, i.e. 10 nM mononucleosomes (H3K9me0 and H3K9me3, Epicypher catalog nos. 16-0006 and 16-0315-20) constant in binding buffer [20 mM Hepes (pH 7.5), 4 mM tris, 80 mM KCl, 0.1% Igepal CA-630, 0.2 mM EDTA, 2 mM DTT, and 10% glycerol]. Samples were incubated at 30 °C for 45 min. A 0.5x tris-borate EDTA 6% acrylamide:bis-acrylamide 37.5:1 gel was pre-run at RT for at least 1 hour at 75 V. Reactions were loaded on the gel and ran under the same conditions for 3 hours. Gels were poststained for 2 hours with polyacrylamide gel electrophoresis (PAGE) GelRed DNA stain (Biotium) and imaged using a Typhoon Imager. The unbound nucleosome band was quantified using ImageJ, and binding curves were fit using nonlinear regression (Prism 10).

## Reporting summary

Further information on research design is available in the Nature Portfolio Reporting Summary linked to this article.

# Data availability

The sequencing data generated in this study have been deposited in NCBI's Gene Expression Omnibus (GEO) and are accessible through GEO Series accession number GSE248428. The mass spectrometry

proteomics data generated in this study have been deposited to the ProteomeXchange Consortium via the PRIDE partner repository with the dataset identifier PXD047651[70,71]. The Alphafold-2 multimer models are available in ModelArchive (www.modelarchive.org) with the accession codes ma-dl26b, ma-n4bjg, ma-01x2c, and ma-j3lwp. Source data are provided with this paper.

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

## Acknowledgements

We thank Danesh Moazed for sharing the fission yeast strains used in this study. We thank Saarang Gopinath for his support in obtaining pre-liminary data during the initial phase of this study. This work was sup-ported by the National Science Foundation (grant no. EF-2316281 to K.R. and J.S.B), National Institutes of Health (grant nos. R35GM137832 to K.R.; T32GM007544 to A.A and A.L.; and T32GM007315 to M.S) and American Cancer Society (grant no. RSG-22-117-01-DMC to K.R.) Molecular gra-phics and analyzes were performed with UCSF Chimera, developed by the Resource for Biocomputing, Visualization, and Informatics at the University of California, San Francisco, with support from NIH P41-GM103311. AlphaFold-Multimer-based in silico models were generated using Cosmic2[64]. We thank the Center for Macromolecular Interactions at Harvard Medical School and the Thermo Fisher Scientific Center for Multiplex Proteomics at Harvard Medical School for their support.

## Author contributions

A.A. and K.R. developed the concept, designed the study, interpreted the findings, and prepared the manuscript. A.A., M.S., A. La., G.R., Z.C., and A. Le. performed experimental studies and analysis. B.K. contributed to experimental studies. M.S., A. Larkin, G.R., Z.C., A. Le, and J.S.B. provided technical support and conceptual advice and edited the manuscript.

## Competing interests

The authors declare no competing interests.
