## [Peer Review File · Nature Communications]

Epigenetic memory is governed by an effector recruitment specificity toggle in Heterochromatin Protein 1Editorial Note: Parts of this Peer Review File have been redacted as indicated to remove third-party material where no permission to publish could be obtained.

REVIEWER COMMENTS

Reviewer #1 (Remarks to the Author):

Ames, Seman et al.

How cells remember their past is a question of general interest. One type of such memory can occur via the inheritance of posttranslational chromatin modifications, whether they be on DNA or histones. In this manuscript, the authors investigate this question using the fission yeast *S. pombe*. By saturation mutagenesis of the effector domain of the HP1 protein, Swi6, the authors identify a single amino acid whose mutation can either increase or decrease the maintenance of HP1-dependent silencing as assayed by a transient tethering assay developed previously. Through an extensive series of experiments ranging from biophysical measurements (both in vitro and in vivo), ChIP experiments, reporter gene assays, AlphaFold-multimer modeling, and quantitative mass spectrometry, the authors come to the conclusion that this amino acid controls the recruitment of two effectors, the anti-silencing factor Epe1 and the ribosome biogenesis nuclease, the rixosome, which has been shown previously to be required for heterochromatin maintenance in *S. pombe* (and interestingly also plays a role in Polycomb silencing in human cells). The data are clear and convincing and of exceptional technical quality. The conclusions follow clearly from the data and are important for our understanding of how HP1 proteins function and evolve. I have suggestions for improvement of the text as follows.

1. I found the title to not reflect the depth and breadth of the paper. Here is an alternative: "Epigenetic memory is governed by an effector recruitment specificity toggle in Heterochromatin Protein 1"
2. line 23: it is not obvious what "auxiliary motif" means in this context. Perhaps replace it with "surface"
3. line 28: replace "selective affect" with "reprogram" to be a little more specific.
4. line 30: Only a single substitution was found that reprograms effector recruitment specificity. How is this "remarkable plasticity?" It seems that either this is a unique surface or that there are other (presumably redundant) mechanisms that enable the evolution of effector specificity or that most effectors are recruited by a shared interaction mechanisms (e.g. the pocket formed by CSD dimerization).
5. line 56: This seems like a straw man argument, especially as the authors discuss 'read-write' (line 84 as an essential part of epigenetic memory. Moreover, other mechanisms have been shown to be important for epigenetic memory (histone deacetylation the rixosome etc). The authors argue that PRC1 association with replication forks and phase separation are not consistent, with the inheritance of modification being a central aspect of memory, which doesn't make sense to me. Such data could easily be accommodated in such models, especially in the context of promoting read-write.
6. line 68: The authors may want to cite Lee et al (NSMB 2023), who described a functional role for reader protein phase separation in heterochromatin assembly fidelity in the yeast *C. neoformans*.
7. line 84 – it seems like this could be discussed earlier in the context of the inheritance of paternal histones.
8. line 99 – again, the authors may want to cite the Lee et al. paper here as it may speak to this issue of the function of phase separation, demonstrating that a reader protein can concentrate a writer protein in vivo.
9. line 112: I don't understand what 'acute' means here. I would delete the word.
10. line 129: "minimal system" There are other fungi that have only 1 HP1 ortholog (rather than two) and in which heterochromatin formation is not coupled to RNAi, so I am unsure what 'minimal' means in this context. I suggest saying "outstanding model system"
11. Lines 142-6: The authors may want to cite Garcia et al., Genes and Development, 2010, for work on the function of SHREC in vivo.
12. line 159: The authors might want to spell out what "proteins involved in ribosome biogenesis" means, namely the rixosome.
13. line 209-10" insert "absence" after "critically dependent on the"

14. line 224-5: It is unclear what "persistent" means in this context. Do the authors mean "heritable"?

15. line 543-52: The authors propose that Epe1 is not a histone H3K9 demethylase. It would be helpful if you the authors could bolster this argument by analysis that demonstrates loss of catalytic and/or cofactor binding residues. If these are conserved, then the authors may want to explain why they believe the protein is not an enzyme as proposed or soften this view.

Reviewer #2 (Remarks to the Author):

In the manuscript "Identification of a novel, tunable interface in 1 the *S.pombe* HP1 protein, Swi6, that underpins epigenetic inheritance." Ames et al describe a new surface residue on the CSD domain of Swi6 that tunes epigenetic silencing behaviors mediated by the protein. In particular, they document how mutations at one particular site, T278, affect the ability of Swi6 to maintain gene silencing following the release of the nucleation signal, in this case TetR-Clr4. They focus on two mutations with opposing phenotypes, gain and loss of maintenance, T278Y and T278K, respectively. Both mutations lead to a loss of interaction with the anti-silencing protein Epe1 and both show increased chromatin residence. Hence the mode of transmitting changes in maintenance requires additional mechanisms and the authors show that the two mutant versions differentially recruit rixosome complexes, among other proteins. In particular, the gain-of maintenance T278Y mutant attracts more rixosome than the WT and they authors show that in the tethered context, maintenance requires the rixosome association with Swi6 and that it is downstream of the central HDAC Clr3. Overall, the authors posit the T278 interface as a tunable surface in HP proteins modulating epigenetic behaviors.

Overall, I really enjoyed reading this paper and I think it provides important insights into the field of epigenetic inheritance. It is of broad significance and will be of interest to a general audience in the epigenetics and chromatin field. The multidisciplinary approach to the problem in particular was very satisfying and I commend the authors on this work. The data are overall of high quality, and the conclusions are broadly supported by the evidence provided. I have mostly minor critiques, but I do feel the lack of insight into loss of maintenance for the key Swi6 T278K mutation needs to be further explored.

Major critique:

Here I really only have one essential suggestion: The paper leaves off without an explanation for the loss of maintenance by Swi6 T278K. This mutant loses interaction with Epe1, which in the WT context would be expected to lead to gain of maintenance. But no additional data hint at why this mutation cannot maintain. Is it potentially a silencing defect that is propagated? For example, this mutation does show elevated silencing in -tet (Figure 1D) but apparently reduced silencing in the epe1 mutant (Fig S1C) and reduced spreading (Figure 3C, -tet). The Mass Spec did not reveal anything of note for beyond Epe1/Bdf2, but insights into the mechanisms of maintenance loss are crucial. Does this mutant may recruit less rixosome than WT?

Minor issues:

1. As mentioned above, the silencing changes (increased/decreased) in Swi6 T278K should be accounted for in the text and explained. I was quite struck by the significant upregulation of transcription in the epe1 mutant I -tet in this context.
2. Small changes in Swi6 protein levels: The authors show a nice qualitative western of the different Swi6 T278 mutants. However, small changes in Swi6 levels can really impact silencing, heterochromatin behaviors are quite sensitive to HP1/Swi6 dosage. I advise the authors to produce quantitative western blots (e.g. via IR fluorescent blots and standard curves) for at least the two major Swi6 mutations T278Y/K and wild type.
3. The 278Y mutant simultaneously shows de-repression at dh, but what looks like increased H3K9me3 over WT at the pericentromere. (Figure 3D and E) Can the authors comment?

4. For the Mass photometry, given the regime of 2.5-20nM, any reference to oligomers should be removed as this is orders of magnitude below the oligomerization Kds.
5. For the nucleosome EMSAs, if it is not possible to infer Kds. As Swi6 does not form one bound complex with the nucleosome, this precludes the plotting of a true isotherm and extraction of a Kd, which depends on the saturation of one bound species (A+B -> A.B). The Kds can be replaced by the empirical K1/2 measure, which can be derived from the measurement of 1-unbound (I notice the authors are measuring unbound, so it should require minimal adjustment, if any).
6. I wonder whether the simplest explanation for increase in vivo chromatin binding is the loss of Epe1 association. Epe1 destabilizes methylated nucleosomes (e.g. Aygun 2013 among other studies), hence likely destabilizes Swi6 on these nucleosomes. Loss of Epe association would stabilize Swi6 on chromatin. Is the regime of Swi6 T278Y /K similar to Swi6 WT in epe1 mutants?

Very minor:

1. S Figure 6 legend includes REF in the text, the authors should fill in the placeholder.
2. The reference on line 140 is the wrong year and appears twice in the references.
3. A few times the author "consistent with previous work" without including a reference. Example line 340 and 353 .

I want to end by encouraging the authors again, this is a nice piece of work. They have done a lot, but I think mechanisms of loss of maintenance by Swi6 T278K are critical and worthwhile piece of the puzzle.

Reviewer #3 (Remarks to the Author):

HP1 family proteins play critical roles in regulating heterochromatin integrity by recruiting diverse factors. To further explore HP1 function, the authors mutagenized the *S. pombe* HP1 homologue, Swi6, and identified interesting mutations at T278. These mutations included some gain-of-function (exemplified by Swi6-Y) and some loss-of-function (exemplified by Swi6-K) variants in regulating heterochromatin inheritance. The authors further demonstrated that these mutations affect distinct protein-protein interactions, while leaving Swi6 core functions such as dimerization, nucleosome binding, unaffected. The authors propose that even minor changes in Swi6 can result in significant changes in epigenetic inheritance, suggesting a potential role in HP1 protein evolution.

The results highlight that regions outside of the HP1 dimer interface contribute to interactions of HP1 with other proteins, thereby influencing functional outcomes. The findings are of broad interest to the epigenetics field. The characterization of the two classes of mutations is extensive, the data presented is of high-quality, and the data support the conclusions. Therefore, I only have minor concerns.

1. The primary focus of the manuscript revolves around Swi6-Y and Swi6-K, with a more emphasis on Swi6-Y. For Fig. S3A, only data on Swi6-K is presented. It would be interesting to determine whether Swi6-Y also reduce interaction with Epe1 in this assay.
2. Regarding the structure prediction of Swi6-Epe1 interaction, is T278 located at an interaction interface? Would the substitutions T278Y or T278K be anticipated to impact this interaction?
3. The authors cited that the *grc3-V70M* mutation is a separation of function mutation that disrupts rixosome-Swi6 interaction. Is the V70M mutation expected to disrupt the Swi6-Grc3 interaction based on structural prediction?
4. Have the authors tried to predict Swi6-Grc3 interaction using Swi6-Y? It is possible that Swi6-Y could induce a more favorable configuration for interaction in the flexible region?
5. One of the main claims in the manuscript is the ability of HP1 protein to evolve and acquire new functions, based on the gain-of-function mutation Swi6-Y. Has there been any observation of organism carrying Swi6-Y variant display enhanced epigenetic inheritance compared to similar organisms with

wild type HP1?

6. Is there any explanation for the phenotypes associated with Swi6-K?

We thank the reviewers for their insightful and constructive feedback highlighting the innovation and impact of our findings. We are grateful that the reviewers deemed our work a strong candidate for the broad readership of Nature Communications. We fully acknowledged their critiques and suggestions, which were invaluable during the process of revising our manuscript.

We have carefully addressed the major suggestions, which are broadly summarized here.

Content	Suggested by	Presented in
Title Change	Reviewer #1	Main text
Technical changes, language corrections	Reviewer #1 Reviewer #2 Reviewer #3	Main text: lines 22, 26, 47, 77, 101, 116, 139, 192, 308, 329, 505, 533
Added new citations	Reviewer #1 Reviewer #2	Main text: lines 64, 90, 129, 156, 319, 333, 533
Molecular basis for Swi6-K maintenance defect: Crb3-TAP ChIP-qPCR	Reviewer #2 Reviewer #3	Figure 5f Main text: line 396
Molecular basis for Swi6-K maintenance defect: Genetic rescues of Swi6-K	Reviewer #2 Reviewer #3	Supplementary Figure 11 Main text: lines 419, 537
Quantitative western blot for Swi6 expression	Reviewer #2	Supplementary Figure 1c

These changes, among other minor edits, are described in below in a point-by-point response:

Reviewer #1 (Remarks to the Author):

Ames, Seman et al.

How cells remember their past is a question of general interest. One type of such memory can occur via the inheritance of posttranslational chromatin modifications, whether they be on DNA or histones. In this manuscript, the authors investigate this question using the fission yeast *S. pombe*. By saturation mutagenesis of the effector domain of the HP1 protein, Swi6, the authors identify a single amino acid whose mutation can either increase or decrease the maintenance of HP1-dependent silencing as assayed by a transient tethering assay developed previously. Through an extensive series of experiments ranging from biophysical measurements (both in vitro and in vivo), ChIP experiments, reporter gene assays, AlphaFold-multimer modeling, and quantitative mass spectrometry, the authors come to the conclusion that this amino acid controls the recruitment of two effectors, the anti-silencing factor Epe1 and the ribosome biogenesis nuclease, the rixosome, which has been shown previously to be required for heterochromatin maintenance in *S. pombe* (and interestingly also plays a role in Polycomb silencing in human cells). The data are clear and convincing and of exceptional technical

quality. The conclusions follow clearly from the data and are important for our understanding of how HP1 proteins function and evolve. I have suggestions for improvement of the text as follows.

We thank the reviewer for their generous and thoughtful feedback relating to changes to the text and adding references. Their feedback was very helpful and we accepted all of their changes and edited the revised submission accordingly.

1. I found the title to not reflect the depth and breadth of the paper. Here is an alternative: “Epigenetic memory is governed by an effector recruitment specificity toggle in Heterochromatin Protein 1”

We appreciate the reviewer’s suggestion regarding a revised manuscript title. Thank you for this very kind suggestion! The revised manuscript title is “*Epigenetic memory is governed by an effector recruitment specificity toggle in Heterochromatin Protein 1*”.

2. Line 23: it is not obvious what “auxiliary motif” means in this context. Perhaps replace it with “surface”

We have changed the text in the manuscript (page 2, line 22) to replace “motif” with “surface”.

3. line 28: replace “selective affect” with “reprogram” to be a little more specific.

We agree with the reviewer’s suggestion and have altered the text (page 2, line 26) accordingly.

4. line 30: Only a single substitution was found that reprograms effector recruitment specificity. How is this “remarkable plasticity?” It seems that either this is a unique surface or that there are other (presumably redundant) mechanisms that enable the evolution of effector specificity or that most effectors are recruited by a shared interaction mechanisms (e.g. the pocket formed by CSD dimerization).

We have removed the word remarkable to ensure that we do not overstate our conclusions. We agree with the reviewer that the CSD dimerization interface and the second novel surface we identified could be redundant to enable the evolution of novel effector specificity. We have altered the text in our manuscript (page 2, line 26) to read as follows:

“We show that relatively minor changes in Swi6 amino acid composition can lead to profound changes in epigenetic inheritance which provides a redundant mechanism to evolve novel effector specificity.”

5. Line 56: This seems like a straw man argument, especially as the authors discuss ‘read-write’ (line 84 as an essential part of epigenetic memory. Moreover, other mechanisms have been shown to be important for epigenetic memory (histone deacetylation the rixosome etc). The authors argue that PRC1 association with replication forks and phase separation are not consistent, with the inheritance of modification being a central aspect of memory, which doesn’t make sense to me. Such data could easily be accommodated in such models, especially in the context of promoting read-write.

We understand the reviewers’ concern with this argument and have removed it from this portion of the text in the introduction section of the revised manuscript (page 3, line 47). The text now reads:

“In this model, modified histones are expected to serve as templates to restore pre-existing histone modification states on newly synthesized DNA¹⁴. Histone-modifying enzymes restore these states by engaging in a process called ‘read-write’ wherein pre-existing histone modifications can recruit enzymes to modify newly incorporated histones¹⁵⁻¹⁹. However, in addition to the enzyme having read-write properties, several non-histone proteins that bind to H3K9me have also been implicated in regulating epigenetic inheritance, in particular by inducing changes in chromatin conformation^{20,21}.”

6. line 68: The authors may want to cite Lee et al (NSMB 2023), who described a functional role for reader protein phase separation in heterochromatin assembly fidelity in the yeast *C. neoformans*.

We thank the reviewer for making us aware of this oversight and are grateful to them for highlighting this exceptional study relating to phase separation in heterochromatin assembly in *C. neoformans*. We agree this study makes important, relevant, and meaningful contributions to our work and have now included this citation (page 4, line 64).

7. Line 84 – it seems like this could be discussed earlier in the context of the inheritance of paternal histones.

We agree with the reviewer’s suggestion to discuss this point earlier. We have moved our discussion of “read-write” in the text (page 3, line 47) and it now reads:

“In this model, modified histones are expected to serve as templates to restore pre-existing histone modification states on newly synthesized DNA¹⁴. Histone-modifying enzymes restore these states by engaging in a process called ‘read-write’ wherein pre-existing histone modifications can recruit enzymes to modify newly incorporated histones¹⁵⁻¹⁹. However, in addition to the enzyme having read-write properties, several non-histone proteins that bind to H3K9me have also been implicated in regulating epigenetic inheritance, in particular by inducing changes in chromatin conformation^{20,21}.”

We have also adjusted the text where we originally discussed this point (page 4, line 77) and the text now reads:

“Further, the aforementioned processes thought to regulate epigenetic inheritance; such as ‘read-write’, oligomerization, and phase separation; accomplish this by co-opting reader domain containing proteins (such as HP1), These observations underscore the critical role that reader domains have in promoting epigenetic inheritance.”

8. Line 99 – again, the authors may want to cite the Lee et al. paper here as it may speak to this issue of the function of phase separation, demonstrating that a reader protein can concentrate a writer protein in vivo.

This is a great suggestion! We have made this change to the text (page 5, line 90).

9. Line 112: I don’t understand what ‘acute’ means here. I would delete the word.

We have removed “acute” from the text (page 5, line 101).

10. line 129: “minimal system” There are other fungi that have only 1 HP1 ortholog (rather than

two) and in which heterochromatin formation is not coupled to RNAi, so I am unsure what 'minimal' means in this context. I suggest saying "outstanding model system"

We have made this change from "minimal" to "outstanding model" (page 6, line 116).

11. Lines 142-6: The authors may want to cite Garcia et al., Genes and Development, 2010, for work on the function of SHREC in vivo.

This work very elegantly places chromatin remodeling as a central feature of epigenetic silencing. We thank the reviewer for the suggestion and have added this citation into the text (page 7, line 129).

12. Line 159: The authors might want to spell out what "proteins involved in ribosome biogenesis" means, namely the rixosome.

We agree and have changed the text accordingly (page 7, line 139). The text now reads:

"Furthermore, we demonstrate that a gain of function substitution leads to specific alterations in protein-protein interactions that are known to regulate epigenetic inheritance; including a loss of interaction with the putative H3K9 demethylase Epe1 and a gain of interaction with the rixosome complex, a central component of ribosome biogenesis which is required for heterochromatin inheritance^{21.68}."

13. line 209-10" insert "absence" after "critically dependent on the"

We have made this correction in the text (page 9, line 192).

14. line 224-5: It is unclear what "persistent" means in this context. Do the authors mean "heritable"?

In this context, we used the word persistent to describe the maintenance phenotypes associated with each our mutants. We envisioned that the word "persistent" would allow readers to understand that the Swi6-Y and Swi6-K variants exhibit a permanent, irrevocable gain or loss of maintenance respectively.

15. line 543-52: The authors propose that Epe1 is not a histone H3K9 demethylase. It would be helpful if you the authors could bolster this argument by analysis that demonstrates loss of catalytic and/or cofactor binding residues. If these are conserved, then the authors may want to explain why they believe the protein is not an enzyme as proposed or soften this view.

This is an important question relating to how Epe1 regulates heterochromatin in *S.pombe*. As the reviewer has correctly stated there are several residues involved in co-factor binding (iron and α -ketoglutarate) that are conserved in *S.pombe* (except for Tyr 370 which is typically a histidine in all active demethylases characterized to date).

All previous attempts to reconstitute Epe1 activity including our previous work (Raiymbek et al., *eLife*, 2020, doi: 10.7554/eLife.53155) were unable to measure H3K9 demethylation *in vitro*. This does not rule out the possibility that Epe1 may function as a demethylase with a different set of substrates or in complex with other accessory factors. Nevertheless, in Raiymbek et al., *eLife*, 2020, we were able to demonstrate a non-enzymatic function that could explain how Epe1 disrupts heterochromatin formation. Most strikingly, we discovered that catalytic residue

mutations led to the loss of interaction with its primary binding partner, Swi6. Furthermore, we showed that reduced Epe1 recruitment leads to increased binding of the histone deacetylase, Clr3. Tethering Clr3 at an ectopic locus led to epigenetic inheritance in the presence of Epe1.

We have added this citation to our manuscript (page 28, line 533) and hope that this adds support to a potential non-enzymatic function for Epe1. We reiterate that these observations do not rule out the possibility that Epe1 may function as an enzyme under conditions that we have not explicitly tested but based on our work and work from other labs for example, Sorida et al., 2019 doi: 10.1371/journal.pgen.1008129), we feel it is appropriate to refer to Epe1 as a putative H3K9 demethylase.

To soften our claim, we have omitted the following sentence: “Our results propose a new non-catalytic function for Epe1, a putative demethylase with no known enzymatic activity despite having structural similarities to JmjC containing histone demethylases.” The revised text now reads (page 28, line 533):

*“We have previously shown that substitution mutations, which are thought to affect Epe1 catalytic activity, lead to a loss of a direct interaction between Epe1 and Swi6⁶⁶. The trade-off in protein-protein interactions that we observed in the case of Swi6-Y lends additional support to a model where Epe1 regulates epigenetic inheritance by attenuating the interaction between Swi6 and heterochromatin maintenance enhancers such as the rixosome (**Figure 6**).”*

Reviewer #2 (Remarks to the Author):

In the manuscript "Identification of a novel, tunable interface in 1 the *S.pombe* HP1 protein, Swi6, that underpins epigenetic inheritance." Ames et al describe a new surface residue on the CSD domain of Swi6 that tunes epigenetic silencing behaviors mediated by the protein. In particular, they document how mutations at one particular site, T278, affect the ability of Swi6 to maintain gene silencing following the release of the nucleation signal, in this case TetR-Clr4. They focus on two mutations with opposing phenotypes, gain and loss of maintenance, T278Y and T278K, respectively. Both mutations lead to a loss of interaction with the anti-silencing protein Epe1 and both show increased chromatin residence. Hence the mode of transmitting changes in maintenance requires additional mechanisms and the authors show that the two mutant versions differentially recruit rixosome complexes, among other proteins. In particular, the gain-of maintenance T278Y mutant attracts more rixosome than the WT and they authors show that in the tethered context, maintenance requires the rixosome association with Swi6 and that it is downstream of the central HDAC Clr3. Overall, the authors posit the T278 interface as a tunable surface in HP proteins modulating epigenetic behaviors.

Overall, I really enjoyed reading this paper and I think it provides important insights into the field of epigenetic inheritance. It is of broad significance and will be of interest to a general audience in the epigenetics and chromatin field. The multidisciplinary approach to the problem in particular was very satisfying and I commend the authors on this work. The data are overall of high quality, and the conclusions are broadly supported by the evidence provided. I have mostly minor critiques, but I do feel the lack of insight into loss of maintenance for the key Swi6 T278K mutation needs to be further explored.

We thank the reviewer for providing very thoughtful feedback and highlighting specific areas that could benefit from new experiments and also the need for additional controls. Specifically, their comments led us to test models for how Swi6-K exhibits a loss of maintenance and provide a mechanistic basis for these observations.

Major critique:

Here I really only have one essential suggestion: The paper leaves off without an explanation for the loss of maintenance by Swi6 T278K. This mutant loses interaction with Epe1, which in the WT context would be expected to lead to gain of maintenance. But no additional data hint at why this mutation cannot maintain. Is it potentially a silencing defect that is propagated? For example, this mutation does show elevated silencing in -tet (Figure 1D) but apparently reduced silencing in the epe1 mutant (Fig S1C) and reduced spreading (Figure 3C, -tet). The Mass Spec did not reveal anything of note for beyond Epe1/Bdf2, but insights into the mechanisms of maintenance loss are crucial. Does this mutant may recruit less rixosome than WT?

We thank the reviewer for this question relating to the molecular basis for the persistent loss of maintenance phenotype in Swi6-K. To summarize- in our initial submission, our proteomics data captured a substantial difference in rixosome binding (Grc3, Las1, Rix1, Crb3) between Swi6-Y and Swi6-WT. These results suggested that increased rixosome binding could account for the gain of maintenance in the case of the Swi6-Y variant. However, these mass spectrometry measurements did not reveal any differences between Swi6-K and Swi6-WT. In response to this question by Reviewer 2, we performed two new experiments that provided a compelling rationale for the loss of maintenance in Swi6-K:

1) We considered that mass spectrometry may not have the dynamic range to detect small changes in rixosome binding in the Swi6-K mutant or that the Swi6-K mutant may exhibit differences in rixosome binding but these changes may be restricted to the chromatin bound fraction. To address these possibilities, we performed chromatin immunoprecipitation followed by qPCR (ChIP-qPCR) to detect the chromatin bound fraction of Crb3. We added a tandem affinity purification tag (TAP) to the C-terminus of Crb3 in strains containing Swi6-WT, Swi6-Y or Swi6-K (**Figure 5f**). We observed a significant decrease in Crb3-TAP occupancy in the case of Swi6-K relative to Swi6-WT at the pericentromeric repeats (*dg*) and our reporter locus (*SPCC330.06c*), respectively. As expected, we observed a notable increase in Crb3-TAP binding in Swi6-Y expressing cells relative to wild-type. These results suggest that decreased rixosome binding to Swi6-K leads to the persistent loss of maintenance phenotype.

We have added these new results to the revised manuscript (page 22, line 396):

"Our mass spectrometry data provided a straightforward explanation for enhanced maintenance in the case of Swi6-Y. However, the molecular basis of Swi6-K defective maintenance remains unclear. We considered the possibility that a change in rixosome binding in Swi6-K is either too small for sensitive detection using mass spectrometry or that these changes correspond only to the rixosome chromatin bound fraction. To test this, we fused the Crb3 subunit of the rixosome with a C-terminal TAP tag and performed ChIP-qPCR to measure its occupancy at the pericentromeric repeats (*dg*) and the 10XtetO-*ade6+* reporter locus (*SPCC.330.06c*) (**Figure 5f**). We detected a small but significant decrease in rixosome binding at both loci in Swi6-K relative to Swi6-WT. Consistent with our mass spectrometry findings, we observe a significant increase in rixosome binding at the pericentromeric repeats and reporter gene locus in Swi6-Y. These results suggest that the persistent loss of maintenance observed in Swi6-K is likely due to reduced rixosome binding at heterochromatin (**Figure 5f**). These observations also explain why deleting *Epe1* cannot rescue the maintenance defect in Swi6-K (**Figure 1f**)."

2) As we previously noted, the defective maintenance in Swi6-K is likely due to changes in protein interactions that were not captured by our mass spectrometry experiment. Hence, we hypothesized that there may be compensatory mutations in the Swi6-K sequence context that could rescue maintenance. To identify such mutations, we utilized a PCR based targeted saturation mutagenesis approach described in **Supplementary Figure 1a** to generate a library of variants in residues proximal to Swi6-K (residues: 268-277, 279-302). The premise of this screen was to recover variants that exhibited a gain of maintenance resulting in colonies that

turned red in +tet medium (Swi6-K has defective maintenance that leads to white colonies in +tet medium).

This screen was successful and revealed a series of substitutions at D283 (H, T, S, R, E) that could rescue the maintenance defect in a Swi6-K background (**Supplementary Figure 11b**). A previous study has shown that this region of Swi6 (residues 282 to 284) may be involved in binding to a pro-maintenance histone chaperone called FACT (Takahata et al., 2021, Holla et al. Cell, 2020). FACT is a heterodimeric complex consisting of two proteins-Spt16 and Pob3. *In vitro* studies showed that alanine substitutions within the unstructured loop region (which includes D283A) led to a loss of FACT binding. We envision that the compensatory D283 substitutions we identified in the Swi6-K background could lead to an increase in FACT complex recruitment. These data support our model for how subtle yet specific variations in amino acid composition in HP1 proteins leads to highly divergent effects on epigenetic inheritance. These observations are also consistent with our data where tethering another pro-maintenance factor, the histone deacetylase Clr3 at an ectopic site, rescued defective maintenance in the Swi6-K background (**Supplementary Figure 11a**).

We have added these new results to the revised manuscript:

(page 24, line 419)

“Genetic rescue of Swi6-K heterochromatin maintenance defects

*Previous work has shown that targeting the histone deacetylase Clr3 to heterochromatin (either by tethering Clr3 or by fusing chromodomains) is sufficient for epigenetic inheritance despite the presence of Epe1^{86,85}. The HDAC activity of Clr3 reduces histone turnover, a characteristic feature of heterochromatin that is thought to promote epigenetic inheritance⁸⁸. We wanted to test if tethering Clr3 is sufficient to rescue defective maintenance in swi6-K expressing cells. We expressed a Gal4-Clr3 fusion protein in strains containing two orthogonal DNA binding sequences, i.e. 10XUAS sites for Gal4 binding and 10XtetO sites for TetR binding- both of which are placed upstream of the ade6+ reporter gene (Supplementary Figure 11a). Despite Epe1 being present, we observed robust maintenance of ade6+ silencing, with cells appearing red or sectoring when plated on +tet media (Supplementary Figure 11a, swi6-WT, gal4-clr3). This process is critically dependent on Swi6 since both establishment and maintenance were eliminated in cells lacking Swi6 (**Supplementary Figure 11a**, swi6Δ, gal4-clr3). Interestingly, tethering Clr3 rescued the Swi6-K maintenance defect since we observed both successful establishment (red colonies, -tet) and maintenance (red or sectoring colonies in +tet), although maintenance was not nearly as robust as what we observed in swi6-WT cells (**Supplementary Figure 11a**). Furthermore, targeting Clr3 could not bypass the requirement for the rixosome interaction in heterochromatin maintenance. In cells expressing grc3-V70M, tethering Clr3 failed to produce red or sectoring colonies when cells were plated on +tet-containing medium (**Supplementary Figure 11a**). Hence, Clr3-mediated histone deacetylation can compensate for defective heterochromatin maintenance in the case of swi6-K but not in the case of grc3-V70M.*

*We envisioned that an alternative approach to detect the molecular basis for defective maintenance in Swi6-K cells would be to identify other residues in Swi6 that could suppress this defect. These compensatory mutations could potentially restore protein interactions that are lost in Swi6-K cells. To identify such mutations, we utilized a PCR based targeted saturation mutagenesis approach as previously described (**Supplementary Figure 1a**) to generate a library of variants in residues that are proximal to Swi6-K (residues: 268-277, 279-302). This screen revealed substitutions at D283 to histidine (H), threonine (T), serine (S), arginine (R), and glutamic acid (E) that could rescue the maintenance defect in the Swi6-K background,*

indicated by red or sectored colonies on YE+tet plates (**Supplementary Figure 11b**). This residue, D283, falls within a region of Swi6 involved in binding to the histone chaperone complex, FACT⁸⁷. Collectively, our observations suggest that the Swi6-K maintenance defect is likely mediated by changes in protein interactions, correlated with decreased rixosome binding.”

(page 28, line 537)

“Our findings suggest that the molecular basis for the loss of maintenance in the case of Swi6-K is likely also mediated by changes in protein-protein interactions. We can rescue the maintenance defect observed in Swi6-K two ways: through D283 compensatory mutations or by tethering Clr3 at the ectopic locus (**Supplementary Figure 11**). Given D283 has been previously implicated in FACT complex binding, our findings suggest these compensatory D283 (H, T, R, S and E) substitutions may lead to a gain in FACT binding⁸⁷. Additionally, Clr3 tethering has been shown to bypass essential factors in heterochromatin inheritance, including the rescue of reduced H3K9me density, increased histone turnover, and impaired heterochromatin positioning at the nuclear periphery⁸⁵. Therefore, the increased recruitment of "maintenance factors" such as Clr3 or FACT can rescue the defective maintenance in Swi6-K cells.”

Minor issues:

1. As mentioned above, the silencing changes (increased/decreased) In Swi6 T278K should be accounted for in the text and explained. I was quite struck by the significant upregulation of transcription in the *epe1* mutant I -tet in this context.

We agree with the reviewer that this is an intriguing observation. We hope that the new data we have added to the manuscript addresses the defect in Swi6-K associated heterochromatin.

2. Small changes in Swi6 protein levels: The authors show a nice qualitative western of the different Swi6 T278 mutants. However, small changes in Swi6 levels can really impact silencing, heterochromatin behaviors are quite sensitive to HP1/Swi6 dosage. I advise the authors to produce quantitative western blots (e.g. via IR fluorescent blots and standard curves) for at least

the two major Swi6 mutations T278Y/K and wild type.

The reviewer brings up a valid point that Swi6 silencing is dosage-dependent and small changes in Swi6 levels can affect silencing. Therefore, we have performed the proposed fluorescence as a quantitative readout for western blots (**Supplementary Figure 1c**). We used the linear range of the fluorescence signal upon secondary antibody binding to measure Swi6 protein levels in triplicate across the primary strains used in this manuscript- Swi6-WT, Swi6-Y and Swi6-K. We observed no significant difference in protein levels across the three strains, supporting our model that changes in protein-protein interactions, and not dosage, affect the observed maintenance phenotypes.

3. The 278Y mutant simultaneously shows de-repression at dh, but what looks like increased H3K9me3 over WT at the pericentromere. (Figure 3D and E) Can the authors comment?

This is an intriguing observation but is also consistent with what often happens at pericentromeric repeats in *epe1Δ* strains. Silencing within the pericentromeric repeats is RNAi dependent. As a result, transcription during each S-phase is required for small RNA production and subsequent heterochromatin establishment (Chen et al., Nature, 2008- doi: 10.1038/nature06561, Kloc et al., Current Biology, 2008- doi: 10.1016/j.cub.2008.03.016). Previous studies have proposed that Epe1 is required to initiate pericentromeric silencing by stimulating transcription within repeat regions through RNA polymerase II recruitment and this process cannot happen in *epe1Δ* cells (Zofall et al., Molecular Cell, 2006- doi: 10.1016/j.molcel.2006.05.010, Bao et al., Genes and Development, 2018- doi: 10.1101/gad.318030.118., Asanuma et al., Genes and Development, 2022- doi: 10.1101/gad.350129.122.). Since, Swi6-Y exhibits a loss of interaction with Epe1, we envision that these mutants are likely to display phenotypes that resemble *epe1Δ* cells. Most notably, the Allshire lab (Trewick et al., EMBO J., 2007- doi:10.1038/sj.emboj.7601892) showed that *epe1Δ* cells produced an unstable heterochromatin silencing phenotype wherein cells stochastically silence or activate reporter gene expression. The net effect of such heterogeneity in silencing phenotypes is increased transcription and a defect in transcription gene silencing at the pericentromeric repeat sequences despite the loss of Epe1 leading to increased H3K9me.

4. For the Mass photometry, given the regime of 2.5-20nM, any reference to oligomers should be removed as this is orders of magnitude below the oligomerization Kds.

We thank the reviewer for highlighting this technical issue and we have removed the reference to oligomers in the text pertaining to mass photometry (page 17, line 308). The text now reads:

“Mass Photometry (MP) is a single-molecule approach that uses light to detect the number and molar mass of unlabeled molecules in dilute samples and, given its measurement range, we expected to detect mass differences between Swi6 monomers and dimers⁷⁶.”

5. For the nucleosome EMSAs, it is not possible to infer Kds. As Swi6 does not form one bound complex with the nucleosome, this precludes the plotting of a true isotherm and extraction of a Kd, which depends on the saturation of one bound species (A+B → A.B). The Kds can be replaced by the empirical K_{1/2} measure, which can be derived from the measurement of 1-unbound (I notice the authors are measuring unbound, so it should require minimal adjustment, if any).

We thank the reviewer for bringing attention this oversight. We have replaced our reported binding constants to reflect the empirical K_{1/2} measurement and confirm that our analysis does reflect the "1-unbound population" (**Figure 4c**). We have also made the corresponding corrections to the text (page 18, lines 329). The text now reads:

*“After fitting our concentration-dependent binding assays, we determined that Swi6-WT binds to H3K9me3 mononucleosomes with an apparent K_{1/2} of ~50nM, which was very similar to the K_{1/2} for Swi6-Y binding to H3K9me3 nucleosomes (~39nM). In addition, both Swi6-WT and Swi6-Y bind to H3K9me3 mononucleosomes with similar specificity (2.4-fold for Swi6-WT and 2.3-fold for Swi6-Y) (**Figure 4a-c**).”*

6. I wonder whether the simplest explanation for increase in vivo chromatin binding is the loss of Epe1 association. Epe1 destabilizes methylated nucleosomes (e.g. Aygun 2013 among other studies), hence likely destabilizes Swi6 on these nucleosomes. Loss of Epe association would stabilize Swi6 on chromatin. Is the regime of Swi6 T278Y /K similar to Swi6 WT in epe1 mutants?

The reviewer proposes an attractive model in which the loss of Epe1 binding is sufficient to explain increased Swi6 chromatin occupancy as observed in the case of Swi6-Y and Swi6-K. According to this model, the loss of Epe1 binding may promote Swi6 binding to H3K9me nucleosomes which could be interrupted by Epe1 associated H3K9 demethylase activity. However, in previous studies where we measured Swi6 dynamics in *epe1Δ* cells, we were unable to observe any change in Swi6 chromatin occupancy (Biswas et al., Science Advances, 2022- doi: 10.1126/sciadv.abk0793). Hence, we think there must be other changes that occur in the Swi6-Y and Swi6-K variants that affects their chromatin occupancy. NMR studies have shown that the Swi6 CSD domain binds to the nucleosome core and these studies have detected substantial chemical shifts within the Swi6 Thr278 associated beta sheet interface (Sanulli et al., Nature 2019- doi: 10.1038/s41586-019-1669-2). Although more evidence is needed, these observations raise the possibility that substitutions within the beta sheet interface in Swi6 (which includes Thr 278) could lead to the enhanced chromatin occupancy due to CSD-nucleosome interactions.

[redacted]

Very minor:

1. S Figure 6 legend includes REF in the text, the authors should fill in the placeholder

We have corrected this in the figure legend and included the corresponding reference.

2. The reference on line 140 is the wrong year and appears twice in the references.

We have corrected the year and removed the duplicate reference.

3. A few times the author “consistent with previous work” without including a reference. Example line 340 and 353 .

We thank the reviewer for highlighting this. We have now included citations that highlight previous work in the revised manuscript. We made the following changes to the text:

(page 7, line 156)

*“Consistent with previous work, deleting the eraser of H3K9me, Epe1 (epe1Δ), is necessary to maintain ade6+ silencing, as indicated by the appearance of red and sectorized colonies that persist on +tetracycline-containing medium in contrast to white colonies in epe1+ cells (**Figure 1b**, epe1+ versus epe1Δ, +tet)¹⁸.”*

(page 18, line 319)

“Consistent with previous work, Swi6-WT dimerizes with an apparent K_{dim} of 0.38 nM. We did not observe a significant change in dimerization affinity in Swi6-Y and Swi6-K with apparent K_{dim} values being 0.27 nM and 0.20 nM, respectively⁵⁸.”

(page 18, line 333)

“Consistent with previous studies, we confirmed the complete loss of specificity for H3K9me3 binding in control experiments measuring Swi6 L315E (Figure 4c)⁷⁸.”

I want to end by encouraging the authors again, this is a nice piece of work. They have done a lot, but I think mechanisms of loss of maintenance by Swi6 T278K are critical and worthwhile piece of the puzzle.

We are grateful to the reviewer for their suggestions and happy to hear they enjoyed reading our manuscript. We hope the changes we made to the manuscript assuage any remaining concerns they may have about the impact of our study.

Reviewer #3 (Remarks to the Author):

HP1 family proteins play critical roles in regulating heterochromatin integrity by recruiting diverse factors. To further explore HP1 function, the authors mutagenized the *S. pombe* HP1 homologue, Swi6, and identified interesting mutations at T278. These mutations included some gain-of-function (exemplified by Swi6-Y) and some loss-of-function (exemplified by Swi6-K) variants in regulating heterochromatin inheritance. The authors further demonstrated that these mutations affect distinct protein-protein interactions, while leaving Swi6 core functions such as dimerization, nucleosome binding, unaffected. The authors propose that even minor changes in Swi6 can result in significant changes in epigenetic inheritance, suggesting a potential role in HP1 protein evolution.

The results highlight that regions outside of the HP1 dimer interface contribute to interactions of HP1 with other proteins, thereby influencing functional outcomes. The findings are of broad interest to the epigenetics field. The characterization of the two classes of mutations is extensive, the data presented is of high quality, and the data support the conclusions. Therefore, I only have minor concerns.

We greatly appreciate the reviewer highlighting how our work is of broad interest to field of epigenetics. We are very grateful to them for their thoughtful feedback that helped enhance the clarity of our manuscript. Most notably, their comments also led us to test models for how Swi6-K exhibits a loss of maintenance and provide a mechanistic basis for these observations.

1. The primary focus of the manuscript revolves around Swi6-Y and Swi6-K, with a more emphasis on Swi6-Y. For Fig. S3A, only data on Swi6-K is presented. It would be interesting to determine whether Swi6-Y also reduce interaction with Epe1 in this assay.

We agree with the reviewer that it would be ideal to include Swi6-Y in this assay. However, we did not prioritize this experiment because we interrogated the Epe1-Swi6 interaction in both mutants through two other orthogonal approaches: Epe1 co-immunoprecipitation (**Figure 3b**) and Quantitative AP-MS (**Figure 5a-b**). Additionally, the Swi6-Y silencing phenotypes we observe are consistent with a loss of interaction with Epe1. Since Swi6-K unusually exhibits a persistent loss of maintenance (an unexpected finding), we thought it was necessary to validate the result *in vitro*. We would like to respectfully highlight the significant challenge and resource-intensive nature of this additional *in vitro* experiment, as it requires the generation and purification of new proteins using an insect cell purification system. We hope the reviewer will agree (given current resource constraints) that this experiment would add only incremental value to the overall conclusions of our manuscript.

2. Regarding the structure prediction of Swi6-Epe1 interaction, is T278 located at an interaction interface? Would the substitutions T278Y or T278K be anticipated to impact this interaction?

In the structural prediction of Swi6-Epe1, the region of Epe1 that would interact with Thr 278 is mostly disordered and predicted with low confidence. However, upon Swi6 binding, a helix appears within Epe1 proximal to the beta sheet surface where Thr 278 is located. Given its location within this novel surface, we do expect that Swi6-Y and Swi6-K substitutions directly impact Epe1 binding (also supported by our co-IP and *in vitro* binding measurements in **Figure 3b** and **Supplementary Figure 3a**).

3. The authors cited that the grc3-V70M mutation is a separation of function mutation that disrupt rixosome-Swi6 interaction. Is the V70M mutation expected to disrupt the Swi6-Grc3 interaction based on structural prediction?

The Grc3 V70M mutation was identified in previous studies based on Grc3 having a putative PxVxL motif. Additional biochemical studies showed a loss of the Swi6-rixisome interaction using quantitative AP-MS (Shipkovenska et al., eLife, 2020- doi: 10.7554/eLife.54341). Hence, our AlphaFold2 Multimer structural predictions support these biochemical studies since the Val 70 residue is part of the PxVxL motif that interacts with the Swi6 CSD dimer interface (**Supplementary Figure 9c**).

4. Have the authors tried to predict Swi6-Grc3 interaction using Swi6-Y? It is possible that Swi6-Y could induce a more favorable configuration for interaction in the flexible region?

AlphaFold2 Multimer performs poorly when predicting structural changes arising from point mutations in protein complexes (Yang et al, Signal Transduction and Targeted Therapy 2023-

doi: <https://doi.org/10.1038/s41392-023-01381-z>). Although we speculate that Swi6 variants could induce more favorable interactions, we refrained from using AF-M to justify this claim as the resulting models using Swi6-Y or Swi6-K (single amino acid substitutions) could be incorrect or misleading.

5. One of the main claims in the manuscript is the ability of HP1 protein to evolve and acquire new functions, based on the gain-of-function mutation Swi6-Y. Has there been any observation of organism carrying Swi6-Y variant display enhanced epigenetic inheritance compared to similar organisms with wild type HP1?

The reviewer has brought up an important discussion point that we have considered throughout this work. We did not find a genome within sequenced *Schizosaccharomyces* species which have a Swi6-Y or other amino acid substitution. Therefore, we expect that Thr 278 must have a crucial in regulating Swi6-mediated epigenetic inheritance in these organisms. Any other substitution produced persistent gain or loss of epigenetic inheritance implying a loss of regulation. In our alignments, we did observe that the second HP1 protein Chp2 has an alanine (A) substitution in place of Threonine (T) in *Schizosaccharomyces japonicus* (**Supplementary Figure 12**). Further studies will be needed to determine if this amino acid change impacts Chp2 functions in *S.japonicus*.

One reason why we may not have detected variants that have altered heterochromatin maintenance functions is that residues beyond Thr 278 (such as D283 that we identified in new data that we have added to the revised manuscript (**Supplementary Figure 11**)) may contribute to such effects. An alternative model is that it is not biologically advantageous for an organism to have unchecked, enhanced or suppressed memory since reversibility is a critical aspect of epigenetic regulation. We have discussed this possibility in our revised manuscript (page 27, line 505). The revised text reads:

“The lineage and variant-specific conservation of this region have important consequences for envisioning what the “ground state” of heterochromatin systems might be in different organisms. Most substitutions, apart from the original Thr 278 residue, led to the persistent gain of epigenetic inheritance, and a subset of charged amino acid substitutions led to a persistent loss of epigenetic inheritance (Figure 1b-c). Both extreme scenarios’ consequences are absolute, with heterochromatin being inflexible and not regulatable. Therefore, our findings suggest Thr 278 and other proximal amino acid residues within the beta-sheet interface in Swi6 and other HP1 proteins contribute to epigenetic plasticity wherein cells can invoke memory depending on changes in their physiology or environment.”

6. Is there any explanation for the phenotypes associated with Swi6-K?

We thank the reviewer for this question relating to the molecular basis for the persistent loss of maintenance phenotype in Swi6-K. To summarize- in our initial submission, our proteomics data captured a substantial difference in rixosome binding (Grc3, Las1, Rix1, Crb3) between Swi6-Y and Swi6-WT. These results suggested that increased rixosome binding could account for the gain of maintenance in the case of the Swi6-Y variant. However, these mass spectrometry measurements did not reveal any differences between Swi6-K and Swi6-WT. In response to this question by Reviewer 2, we performed two new experiments that provided a compelling rationale for the loss of maintenance in Swi6-K:

1) We considered that mass spectrometry may not have the dynamic range to detect small changes in rixosome binding in the Swi6-K mutant or that the Swi6-K mutant may exhibit differences in rixosome binding but these changes may be restricted to the chromatin bound fraction. To address these possibilities, we performed chromatin immunoprecipitation followed by qPCR (ChIP-qPCR) to detect the chromatin bound fraction of Crb3. We added a tandem affinity purification tag (TAP) to the C-terminus of Crb3 in strains containing Swi6-WT, Swi6-Y or Swi6-K (**Figure 5f**). We observed a significant decrease in Crb3-TAP occupancy in the case of Swi6-K relative to Swi6-WT at the pericentromeric repeats (*dg*) and our reporter locus (*SPCC330.06c*), respectively. As expected, we observed a notable increase in Crb3-TAP binding in Swi6-Y expressing cells relative to wild-type. These results suggest that decreased rixosome binding to Swi6-K leads to the persistent loss of maintenance phenotype.

We have added these new results to the revised manuscript (page 22, line 396):

"Our mass spectrometry data provided a straightforward explanation for enhanced maintenance in the case of *Swi6-Y*. However, the molecular basis of *Swi6-K* defective maintenance remains unclear. We considered the possibility that a change in rixosome binding in *Swi6-K* is either too small for sensitive detection using mass spectrometry or that these changes correspond only to the rixosome chromatin bound fraction. To test this, we fused the *Crb3* subunit of the rixosome with a C-terminal TAP tag and performed ChIP-qPCR to measure its occupancy at the pericentromeric repeats (*dg*) and the 10XtetO-*ade6+* reporter locus (*SPCC.330.06c*) (**Figure 5f**). We detected a small but significant decrease in rixosome binding at both loci in *Swi6-K* relative to *Swi6-WT*. Consistent with our mass spectrometry findings, we observe a significant increase in rixosome binding at the pericentromeric repeats and reporter gene locus in *Swi6-Y*. These results suggest that the persistent loss of maintenance observed in *Swi6-K* is likely due to reduced rixosome binding at heterochromatin (**Figure 5f**). These observations also explain why deleting *Epe1* cannot rescue the maintenance defect in *Swi6-K* (**Figure 1f**)."

Figure 5. The *Swi6*-rixosome interaction modulates epigenetic inheritance. (f) ChIP-qPCR measuring *Crb3*-TAP at *tub*, *dg*, and *SPCC330.06c* in indicated genotypes. Error bars indicate SD (N=3 or N=4, replicates plotted for each sample). Mean of each sample was compared to their corresponding *swi6*-WT mean using an unpaired one-tailed t-test (P value cutoff < 0.05) and the significance values are represented as follows P < 0.05 (*), P < 0.01 (**), P < 0.0001 (****).

2) As we previously noted, the defective maintenance in *Swi6-K* is likely due to changes in protein interactions that were not captured by our mass spectrometry experiment. Hence, we hypothesized that there may be compensatory mutations in the *Swi6-K* sequence context that could rescue maintenance. To identify such mutations, we utilized a PCR based targeted saturation mutagenesis approach described in **Supplementary Figure 1a** to generate a library of variants in residues proximal to *Swi6-K* (residues: 268-277, 279-302). The premise of this screen was to recover variants that exhibited a gain of maintenance resulting in colonies that turned red in +tet medium (*Swi6-K* has defective maintenance that leads to white colonies in +tet medium).

This screen was successful and revealed a series of substitutions at D283 (H, T, S, R, E) that could rescue the maintenance defect in a *Swi6-K* background (**Supplementary Figure 11b**). A previous study has shown that this region of *Swi6* (residues 282 to 284) may be involved in binding to a pro-maintenance histone chaperone called FACT (Takahata et al., 2021, Holla et al. Cell, 2020). FACT is a heterodimeric complex consisting of two proteins-Spt16 and Pob3. *In vitro* studies showed that alanine substitutions within the unstructured loop region (which includes D283A) led to a loss of FACT binding. We envision that the compensatory D283 substitutions we identified in the *Swi6-K* background could lead to an increase in FACT complex recruitment. These data support our model for how subtle yet specific variations in amino acid

composition in HP1 proteins leads to highly divergent effects on epigenetic inheritance. These observations are also consistent with our data where tethering another pro-maintenance factor, the histone deacetylase Clr3 at an ectopic site, rescued defective maintenance in the Swi6-K background (**Supplementary Figure 11a**).

We have added these new results to the revised manuscript:

(page 24, line 419)

“Genetic rescue of Swi6-K heterochromatin maintenance defects

Previous work has shown that targeting the histone deacetylase Clr3 to heterochromatin (either by tethering Clr3 or by fusing chromodomains) is sufficient for epigenetic inheritance despite the presence of Epe1^{66,85}. The HDAC activity of Clr3 reduces histone turnover, a characteristic feature of heterochromatin that is thought to promote epigenetic inheritance⁸⁸. We wanted to test if tethering Clr3 is sufficient to rescue defective maintenance in swi6-K expressing cells. We expressed a Gal4-Clr3 fusion protein in strains containing two orthogonal DNA binding sequences, i.e. 10XUAS sites for Gal4 binding and 10XtetO sites for TetR binding—both of which are placed upstream of the *ade6+* reporter gene (Supplementary Figure 11a). Despite Epe1 being present, we observed robust maintenance of *ade6+* silencing, with cells appearing red or sectoring when plated on +tet media (Supplementary Figure 11a, *swi6*-WT, *gal4-clr3*). This process is critically dependent on Swi6 since both establishment and maintenance were eliminated in cells lacking Swi6 (**Supplementary Figure 11a**, *swi6* Δ , *gal4-clr3*). Interestingly, tethering Clr3 rescued the Swi6-K maintenance defect since we observed both successful establishment (red colonies, -tet) and maintenance (red or sectoring colonies in +tet), although maintenance was not nearly as robust as what we observed in *swi6*-WT cells (**Supplementary Figure 11a**). Furthermore, targeting Clr3 could not bypass the requirement for the rixosome interaction in heterochromatin maintenance. In cells expressing *grc3*-V70M, tethering Clr3 failed to produce red or sectoring colonies when cells were plated on +tet-containing medium (**Supplementary Figure 11a**). Hence, Clr3-mediated histone deacetylation can compensate for defective heterochromatin maintenance in the case of *swi6*-K but not in the case of *grc3*-V70M.

We envisioned that an alternative approach to detect the molecular basis for defective maintenance in Swi6-K cells would be to identify other residues in Swi6 that could suppress this defect. These compensatory mutations could potentially restore protein interactions that are lost in Swi6-K cells. To identify such mutations, we utilized a PCR based targeted saturation mutagenesis approach as previously described (**Supplementary Figure 1a**) to generate a library of variants in residues that are proximal to Swi6-K (residues: 268-277, 279-302). This screen revealed substitutions at D283 to histidine (H), threonine (T), serine (S), arginine (R), and glutamic acid (E) that could rescue the maintenance defect in the Swi6-K background, indicated by red or sectoring colonies on YE+tet plates (**Supplementary Figure 11b**). This residue, D283, falls within a region of Swi6 involved in binding to the histone chaperone complex, FACT⁹⁷. Collectively, our observations suggest that the Swi6-K maintenance defect is likely mediated by changes in protein interactions, correlated with decreased rixosome binding.”

(page 28, line 537)

“Our findings suggest that the molecular basis for the loss of maintenance in the case of Swi6-K is likely also mediated by changes in protein-protein interactions. We can rescue the maintenance defect observed in Swi6-K two ways: through D283 compensatory mutations or by tethering Clr3 at the ectopic locus (**Supplementary Figure 11**). Given D283 has been previously implicated in FACT complex binding, our findings suggest these compensatory D283

(H, T, R, S and E) substitutions may lead to a gain in FACT binding⁸⁷. Additionally, *Clr3* tethering has been shown to bypass essential factors in heterochromatin inheritance, including the rescue of reduced H3K9me density, increased histone turnover, and impaired heterochromatin positioning at the nuclear periphery⁸⁵. Therefore, the increased recruitment of "maintenance factors" such as *Clr3* or FACT can rescue the defective maintenance in *Swi6-K* cells."

REVIEWERS' COMMENTS

Reviewer #1 (Remarks to the Author):

The authors have adequately addressed my comments. I have no further concerns.

Given the paper's exceptional significance to the field, I wonder why this paper is not being published in Nature Structural and Molecular Biology rather than Nature Communications.

Reviewer #2 (Remarks to the Author):

The authors have addressed most of my concerns. For the mechanism of swi6-K inheritance loss, they provide some data that rixosome recruitment is somewhat compromised. From the Figure 5F panel, this looks quite minor. Could the authors spell out in the manuscript the fold change and also compare that to the much more robust change in WT- swi6-Y? It would be important to have that information.

The authors provide additional data that mutants within swi6-K can restore inheritance. They speculate that this might modulate FACT recruitment. It's a shame they leave it somewhat hanging here, as this in itself does provide evidence of whether FACT recruitment is involved. It also does not really answer more directly what the mechanism underlying the swi6-K defect might be; we are still left with this minor grb3 recruitment defect. Nonetheless, I do feel the authors have given us at least some initial glimpses as to the mechanism of inheritance loss through swi6-K and thus, with the provision of the quantitative IP data, I would favor publication.

Reviewer #3 (Remarks to the Author):

The revised manuscript has satisfactorily addressed my questions. The additional data also enhanced the manuscript. I support acceptance for publication.

Reviewer #1 (Remarks to the Author):

The authors have adequately addressed my comments. I have no further concerns.

Given the paper's exceptional significance to the field, I wonder why this paper is not being published in Nature Structural and Molecular Biology rather than Nature Communications.

We are delighted about the reviewer's positive endorsement of our manuscript and really appreciate all of their feedback!

Reviewer #2 (Remarks to the Author):

The authors have addressed most of my concerns. For the mechanism of swi6-K inheritance loss, they provide some data that rixosome recruitment is somewhat compromised. From the Figure 5F panel, this looks quite minor. Could the authors spell out in the manuscript the fold change and also compare that to the much more robust change in WT- swi6-Y? It would be important to have that information.

We thank the reviewer for their thorough and constructive feedback of our work. We have updated the text to clearly state the respective fold-changes for Swi6-Y and Swi6-K.

The authors provide additional data that mutants within swi6-K can restore inheritance. They speculate that this might modulate FACT recruitment. It's a shame they leave it somewhat hanging here, as this in itself does provide evidence of whether FACT recruitment is involved. It also does not really answer more directly what the mechanism underlying the swi6-K defect might be; we are still left with this minor grb3 recruitment defect. Nonetheless, I do feel the authors have given us at least some initial glimpses as to the mechanism of inheritance loss through swi6-K and thus, with the provision of the quantitative IP data, I would favor publication.

We thank the reviewer for recognizing the value of this study and we hope to further explore the mechanistic basis for the loss of inheritance in the Swi6-K variant in future studies.

Reviewer #3 (Remarks to the Author):

The revised manuscript has satisfactorily addressed my questions. The additional data also enhanced the manuscript. I support acceptance for publication.

We are very grateful to the reviewer for their feedback and really appreciate their positive endorsement of our manuscript.